# A Spatial Analysis of Urban Streets under Deep Learning Based on Street View Imagery: Quantifying Perceptual and Elemental Perceptual Relationships

**Haozun Sun [1], Hong Xu [1,2,*], Hao He [3], Quanfeng Wei [1], Yuelin Yan [1], Zheng Chen [1], Xuanhe Li [1], Jialun Zheng [1] and Tianyue Li [1]**

[1]   School of Urban Construction, Wuhan University of Science and Technology, Wuhan 430065, China; sunhaozun@wust.edu.cn (H.S.); weiquanfeng@wust.edu.cn (Q.W.); YanLin21@wust.edu.cn (Y.Y.); ZhengChen20020321@wust.edu.cn (Z.C.); lixuanhe@wust.edu.cn (X.L.); jialunzheng@wust.edu.cn (J.Z.); litianyue@wust.edu.cn (T.L.)

[2]   Hubei Provincial Engineering Research Center of Urban Regeneration, Wuhan University of Science and Technology, Wuhan 430065, China

[3]   State Key Laboratory of Software Development Environment, Beihang University, Beijing 100191, China; haohe@buaa.edu.cn

[*]   Correspondence: xuhong@wust.edu.cn

**Abstract:** Measuring the human perception of urban street space and exploring the street space elements that influence this perception have always interested geographic information and urban planning fields. However, most traditional efforts to investigate urban street perception are based on manual, usually time-consuming, inefficient, and subjective judgments. This shortcoming has a crucial impact on large-scale street spatial analyses. Fortunately, in recent years, deep learning models have gained robust element extraction capabilities for images and achieved very competitive results in semantic segmentation. In this paper, we propose a Street View imagery (SVI)-driven deep learning approach to automatically measure six perceptions of large-scale urban areas, including "safety", "lively", "beautiful", "wealthy", "depressing", and "boring". The model was trained on millions of people's ratings of SVIs with a high accuracy. First, this paper maps the distribution of the six human perceptions of urban street spaces within the third ring road of Wuhan (appearing as Wuhan later). Secondly, we constructed a multiple linear regression model of "street constituents–human perception" by segmenting the common urban constituents from the SVIs. Finally, we analyzed various objects positively or negatively correlated with the six perceptual indicators based on the multiple linear regression model. The experiments elucidated the subtle weighting relationships between elements in different street spaces and the perceptual dimensions they affect, helping to identify the visual factors that may cause perceptions of an area to be involved. The findings suggested that motorized vehicles such as "cars" and "trucks" can negatively affect people's perceptions of "safety", which is different from previous studies. We also examined the influence of the relationships between perceptions, such as "safety" and "wealthy". Finally, we discussed the "perceptual bias" issue in cities. The findings enhance the understanding of researchers and city managers of the psychological and cognitive processes behind human–street interactions.

**Keywords:** deep learning; street view images; urban perception; urban planning

## 1. Introduction

Accompanied by the acceleration of industrialization, China's urbanization has developed rapidly despite its low starting point and has made remarkable achievements in a short period. However, behind the rapid development of urbanization is a crude urban development method, which has led to urban problems such as traffic congestion, insufficient public space, and environmental degradation. China's urban construction has

entered a historical stage of transformation and development, with urban construction transforming from crude to stock optimization. The structure of urban spatial quality has become the focus of urban planning and construction, from focusing on the construction of "things" in the city to concentrating on the comfort and quality of life of "people" in the town, and from focusing on the construction of "things" in the city to concentrating on the comfort and happiness of "people" in the city.

In most cities, from 20 to 30 percent of the space is covered by streets. This may surprise some, as many people take streets for granted, but they form an essential part of the urban environment. Even more surprising is that streets are also the main public space in cities, accounting for 80% of the total public space, while other public spaces include parks, squares, and other non-private spaces. As a result, more and more cities and citizens are recognizing the importance of streets as a core element that is essential to the proper functioning of a city and the daily lives of its citizens.

As the fundamental connective tissue of the city, streets not only play a key role in shaping residents' experiences and perceptions of the city's structure, but also influence all aspects of urban life, including mobility, safety, social interaction, environmental sustainability, economic development, and cultural identity. Streets are used for transportation and as public spaces for social interaction and community engagement. Well-designed streets can encourage people to gather and foster a sense of belonging and community. A good streetscape design can influence social behavior, promote interaction, and improve the quality of urban life. Green streets can mitigate the urban heat island effect, improve air quality, reduce stormwater runoff, and enhance the ecological balance.

However, the starting point for many cities is to build modern streets as roads. Streets were originally built to encourage automobile travel and maximize passenger and freight transportation efficiency. This has had several positive and negative effects on cities. On the one hand, the connectivity of roads has dramatically changed the accessibility and freedom of travel for residents, promoting regional economic growth and access to services. On the other hand, the planning and management of the street perspective are focused more on the movement of people and vehicles, and the attention to urban space and life is gradually weakened, creating the phenomenon of sacrificing street life for traffic volume. Based on long-term research and practical experience, it is clear that designing streets as roads does nothing to create livable and sustainable urban environments. Worst of all, behaviors on roads endanger the lives of citizens. Traffic accidents, personal injuries, congestion, and pollution adversely affect human health and well-being, some more drastically and others more slowly.

In recent years, city managers have paid increasing attention to the design and utilization of urban street space, which is an essential conduit for social, economic, and cultural activities. The intricate interactions between the constituent elements of urban streets and human perception must be fully understood to cultivate more harmonious and sustainable urban environments. However, due to the time-consuming nature of field surveys, almost all urban designers and researchers at the time limited themselves to small-scale empirical studies, inductively and deductively discovering links between appearance and social attributes and deepening their logical reasoning rather than exploring principles based on large-scale surveys. Automatic quantitative measurements of the spatial quality in large areas have long faced difficulties and time-varying problems. This situation has prevented planners and designers from scientifically developing rational blueprints for street planning.

As a typical representative of China's modern metropolis, Wuhan condenses the complexity of contemporary urbanization. Wuhan's streets exhibit polarized qualities. In historical contexts, mixed-use developments and waterfront areas contributed positively to the urban fabric. However, issues such as traffic congestion, inadequate pedestrian infrastructure, and preserving traditional neighborhoods must be addressed through rigorous urban planning and development strategies. This paper provides a nuanced exploration of Wuhan's streets and an in-depth analysis of the perceptual responses they elicit. It is

expected to provide insights to guide urban planning and design practices to create more livable and vibrant urban spaces.

The research objectives of this paper are as follows: (1) How can the human perception of street space be automatically measured on a large scale? (2) What are the common constituent elements of street spaces that affect human perception? (3) What are the manifestations of the anthropogenic perception of street space in the study area? This paper introduces a street space quantization method based on SVIs under deep learning. We establish a basic database for street space quality quantification research by acquiring open SVI data [1,2], and, based on the computational power of deep learning [3], we deeply explore the complex relationship between urban street space elements and the perceptions of Wuhan residents and tourists. A street space quality measurement system is constructed based on the dual dimensions of the objective evaluation and subjective recognition of urban street space. Based on the quantitative assessment of Wuhan's street spatial quality, the basic features of the distribution of urban spatial quality are deeply interpreted, and the internal and external driving mechanisms of street spatial quality are analyzed. In addition, one of the critical contributions of this study is to elucidate the subtle weighting relationships between elements in different spaces and the perceptual dimensions they affect. The elucidation of these relationships provides viable insights for urban planners and policymakers, and enhances our understanding of the psychological and cognitive processes behind human interactions with urban spaces.

Finally, this study, at the intersection of urban planning, computer science, and environmental behavior, opens up a novel path to understanding the complex perceptions of urban street spaces. The insights derived from this study have the potential to contribute to a paradigm shift in urban design, fostering urban areas that resonate harmoniously with the diverse perceptual needs of residents.

## 2. Literature Review

In recent years, the concept of human-centered design has received more and more attention from people worldwide, and more and more scholars are paying attention to the fact that the quality of streets, as the most frequent destination for citizens, is important. The study of the spatial quality of streets consists of the objective spatial quality of streets and the subjective psychological perceptions of users. Street research can be traced back to Kevin Lynch's <The image of the city>, which categorizes urban spatial elements as "Path", "Edge", "District", "Node", and "Landmark" according to their characteristics [4]. In 1979, Luroborg Creel categorized streets as urban spatial elements: roads, boundaries, zones, nodes, and markers. Creel identified the street as one of the basic structures of the urban spatial archetype [5]. Streets are divided by the boundaries of buildings and are part of the community. The high-quality development of a city must be distinct from the shaping and upgrading of urban street space, which is the direct display of urban imagery, the central place for citizens' activities, and has a significant impact on the quality of life of urban residents.

### 2.1. Evaluation of Spatial Quality of Streets

Streets are the texture of a city, one of the planar components of a town, and a direct space for residents to perceive their city [6,7]. Streets build up the spatial skeleton of the city and bear the function of urban transportation. The transformation of the attributes of urban streets into transportation functions is closely related to the birth of modern functionalism. Street spaces consist of streets and various elements along these streets, and the spatial structure of the town can be recognized through the plan composed of building groups, squares, and streets.

The Athens Charter is a programmatic document for urban planning in contemporary urban functionalism. The document states that residence is the city's primary function, affirms the streets' attribute as transportation infrastructure, and transforms urban spatial research from the two-dimensional plane to the three-dimensional space. After that, it

lays a solid foundation for studying the street at the three-dimensional spatial level. With the advancement of urbanization, the high-speed development of cities triggers similar problems, such as traffic disorder and environmental safety, and more and more scholars have begun to pay attention to street space from the perspective of human experience. Human-centeredness has shown a comprehensive engagement in urban space research and practice as China's urbanization progresses. It can be found from related studies that, in recent years, the research perspective of urban street space has shifted to the high-quality and sustainable development of cities. Research focusing on human feelings has become more affluent and prosperous and increasingly emphasizes high-quality urban street space. For example, the pedestrian space of urban streets is explored from the perspective of safety [8], the spatial governance of streets is examined from the public health standpoint [9], and the research on street vitality is based on being "people-oriented" [10], etc. Some scholars have also studied the measurement of urban space. Some scholars have also researched the indicators of urban space measurement, such as the street height and width ratio [11], green visibility, sky openness [12], and discount rate, etc. Based on these indicators, some scholars have also studied urban space measurement indicators. Based on these indicators, some scholars have begun to study the feasibility of streetscape images considering the above aspects [13]. Traditional street space research mainly relies on manual measurement and subjective judgment. However, the metrics are gradually maturing, but with streets as the urban texture, it challenging to meet the requirements of large-scale data using manual methods, so combining these with the emerging big data means achieving urban scale research is necessary.

## 2.2. Quantitative Research on Street Space

In recent years, studies on street space evaluation have become more prosperous and more diversified in terms of methodology [14,15]. Traditional research on street spatial quality is mainly carried out using manual measurements and subjective judgment, which brings about specific difficulties for refined street spatial evaluation. With the advancement of research, more and more scholars have introduced quantitative means into the study of spatial assessment [16]. In 1984, the proposal of spatial syntax [17] marked the beginning of quantitative research at the level of a two-dimensional street network. Spatial syntax realizes the quantification and visualization of the accessibility, associativity, and spatial topological relationships of planar streets. Until now, spatial syntax has still been an important research method in urban planning. The two-dimensional analysis technique has been further optimized and developed with the efforts of many scholars and has become the most commonly used technical tool in urban research. With the booming development of computer technology, currently, data acquisition channels at home and abroad are becoming more and more diversified, and the birth of POI (Point of Interest) data has expanded the scale of the quantitative evaluation of urban streets from a macro perspective. Zhang et al. combined POI with a public awareness assignment to extend the accuracy of POI data to the linear space of streets and explored a new method to identify street types in a refined way [18]. Qin et al. quantitatively evaluated the street vitality in tourist cities based on POI, Baidu heat maps, OSM road networks, and other data [19]. Quantitative evaluation requires the realization of quantitative measurement methods, and the current quantitative measurement methods are becoming increasingly diversified with the continuous enrichment of data sources. Geographic Information System (GIS)-based measurements are one of the most commonly used methods in the current research field [20], which generally use road network and POI data as the basis of research. There are also street spatial measurement methods based on 3D modeling, such as the 3D modeling of streets based on 3D virtual reality technology [21]. Although these methods have realized the quantitative study of urban street spaces, they are still limited by data processing. It is challenging to carry out a wide range of refinement studies using them. Deep learning techniques based on convolutional neural networks (CNN) provide new research ideas to solve this dilemma.

Machine Learning [22,23] (ML) was first introduced in 1959 by Samuel [24], a pioneer scholar in artificial intelligence. Machine Learning can find better solutions by analyzing and learning from historical and behavioral data and can simplify many complicated manual tasks. As a branch of machine learning, deep learning [25,26] is now used in various fields, such as face recognition, traffic safety [27], agricultural pests [28], and geographic research [29]. Image segmentation is one of the popular fields of computer vision. Image segmentation means dividing an image (or frame in a video) into several segments according to the objects in the image (or frame) on demand [30]. The tremendous progress in CNNs has laid the foundation for image segmentation supported by deep learning. The goal of traditional image segmentation methods is to segment input images based on semantic information and predict the semantic category of each pixel from a given set of labels. According to different segmentation methods, image semantic segmentation can be categorized into edge-detection-based segmentation methods, region-based segmentation methods, threshold-based segmentation methods, and segmentation methods combining specific theories [31]. At this stage, semantic segmentation network models based on CNNs are constantly advancing and have made significant progress in image segmentation. The common backbone network models include VGGNet [32], ResNet [33], and MobileNet [34], etc. Semantic segmentation can classify pixels according to the labels defined for each element. In recent years, semantic segmentation technology has been utilized in many disciplines, especially remote sensing images. Based on the U-Net model, Hu et al. proposed a multi-scale jump-joining method, introduced the attention mechanism and pyramid pooling module, and solved the problem of fine-grained segmentation in complex backgrounds [35]. This study significantly improved the segmentation accuracy of the less-occupied categories in remote sensing images. Shuai Liu et al. proposed a dual decoupled semantic segmentation network model to cope with the problems of high-frequency information loss and segmentation inaccuracy in remote sensing images [36], which effectively facilitated an improvement in the segmentation accuracy of the feature elements in the pictures. Semantic segmentation effectively alleviates the inefficiency and redundancy of manual identification. It has many applications in bridge corrosion data acquisition [37], building change detection [38], ancient building components research [39], geologic rock chip identification [40], and agricultural farming automation [41].

### 2.3. Spatial Sensing of Streets from a Big Data Perspective

At the beginning of the twenty-first century, the MIT Media Lab launched the "Place Pulse" project, a data collection platform that engages volunteers in experiments on urban perception ratings. By the end of 2016, the MIT "Place Pulse" dataset had collected 1.17 million pairs of 110,988 cityscape images from 81,630 online participants. Inspired by the "place Pulse" dataset and supported by recent advances in deep learning techniques, a large body of research has been conducted to analyze the human perceptions of urban appearance [42,43].

The development and application of big data help to conduct street space research on a larger scale [44,45]. Street view images are an accurate and complete embodiment of street spaces. With the development of Google, Baidu, and other mapping platforms, street view images are more easily accessible. Integrating street view images and deep learning technology opens up a new dimension of quantitative research on urban space, revealing the possibility of breaking through previous limitations, thus bringing about a paradigm shift in research [46]. Taking advantage of this, Zhang et al. applied the U2-net neural network model to analyze streetscape data in order to measure the greening quality of streets in old towns [47]. Xia et al. quantified the sky areas of streets in the context of deep learning based on streetscape imagery [48]. Tang [7] explored a new method for evaluating the spatial visual quality of large-area streets and identifying changes. Zhang [49] proposed improving the environmental quality of streets in old towns by analyzing the microclimate measured data and dynamic simulations and putting forward suggestions for improving the environment of old urban areas. Ye [50] proposed a feasible method for perception-

based quantitative measurements of street visual quality, which usually relies on subjective impressions or sensations. Yan [51] provided an analytical mindset for the planning and construction of streets, which guides the implementation of street-related projects and plans. Sou [52] proposed an Artificial Intelligence and Human Collaborative Evaluation (AIHCE) framework based on human emotions and values and facilitated communication design between designers and stakeholders as a method for street space evaluation. Zhu [53] used multiple linear stepwise regression to model the relationship between street vitality and the intensive land use index at three different times of the day: non-commuter, weekly commuter, and weekend. As an essential part of urban public spaces, streets are the primary carrier of traffic and an essential space for individuals' daily activities (including recreation and communication). Liu [54] highlighted the social characteristics of streets by integrating them into a single vitality index. In order to test the validity of the proposed model, Wan [55] conducted a comparative analysis of the results of public assessment, expert scoring, and model measurements to verify whether the measurements were objective and convincing.

In recent years, studies based on the spatial quality of multi-source data have received increasing attention, and these studies are often based on a multidisciplinary and composite perspective, in order to explore the association between the spatial quantification of street space and social, economic, and psychological aspects. For example, Dai et al. used full convolutional networks (FCN-8s) to segment SVIs to explore whether "blue-green space" was affected by five visual spatial indicators, namely, walkability, closure, openness, imagery, and traffic flow, in influencing the psychology of urban residents [56,57], which explored spatial space's influence mechanism on residents' psychology. The study examined the influence mechanism between space and residents' psychology. Other scholars have paid attention to the relationship between spatial quality evaluation and urban safety, using Google SVI measures to measure the differences between crime and sense of safety [58]. Liu et al. used the Normalized Vegetation Index (NDVI) to study the relationship between neighborhood greening and human psychological health [59]. Some studies have incorporated geographic information dimensions. Deng Yuanyuan et al. combined GIS and SVIs to explore the relationship between the spatial quality of the streets along the Yellow River and the psychological feelings of smoothness of sight corridors, accessibility of space, and space for public activities [60]. In a street spatial quality enhancement discussion, Wang et al. proposed a quality enhancement framework covering quality evaluation, spatial categorization, planning guidelines, and updating time sequences through data enhancement [61].

Although there are now more perspectives from which research the relationship between street space and human perception, the dimensions are relatively homogeneous, and some studies still explore the human psychological perception of space through traditional field research methods, which limits research in related fields on a large scale. In addition, conventional research methods are highly typical, lacking universality, and urban perception is a subjective evaluation that is affected by people's social and cultural backgrounds. Datasets produced through field research methods may be biased when used to predict and assess the urban perceptions in other places. In other words, when we make an accurate perceptual assessment of an area based on traditional research methods, we need to re-obtain a local dataset of the urban perceptions of residents who understand the socio-economic background of the area. Historically, many urban streets were designed primarily for vehicular traffic. Some studies may still prioritize vehicular traffic over the safety and comfort of pedestrians and residents. In addition, in recent years, based on the development of quantitative research tools, scholars have paid more and more attention to quantitative analyses of urban street spaces; however, current studies have yet to refine research on the components of street space and their influencing factors.

Moreover, many existing quantitative research methods need to be more cohesive. Urban street spaces involve various aspects such as transport, urban design, socio-economics, and environmental psychology. There needs to be more interdisciplinary cooperation in

existing research, which may lead to an incomplete understanding of complex urban street spaces. To address these issues in quantifying urban street space perceptions, we propose a spatial perception prediction method based on street images under deep learning, in order to quickly, effectively, and rationally assess urban space perceptions. We selected a high-density urban environment for the study of urban perception assessment. We explored the influence of some urban constituents on human perception and the relationships between perceptions.

## 3. Materials and Methods

### 3.1. Study Area

Wuhan (Figure 1), a sprawling metropolis in central China, possesses a high degree of urban complexity and diversity. Its blend of historical, cultural, and contemporary urban elements provides a unique setting for studying the many facets of human perception in urban environments. Wuhan's high population density and heterogeneous demographic composition epitomize urban society. Wuhan's selection as the study area allows for observing various human perceptions and behaviors. The residents' diverse backgrounds, occupations, and lifestyles constitute a rich perceptual experience. The urban layout of Wuhan demonstrates a fusion of historic neighborhoods and modern development, creating a unique spatio-temporal dynamic that influences human perception. This combination of urban evolution in Wuhan is a product of the interplay between historical continuity and rapid urbanization. Wuhan's cultural significance, including its role as a significant educational, industrial, and commercial center, makes it a compelling site for academic research. In addition, its historical relevance and its strategic location at the confluence of the Yangtze and Han Rivers create unique geographical attributes that can influence human perceptions of the urban landscape.

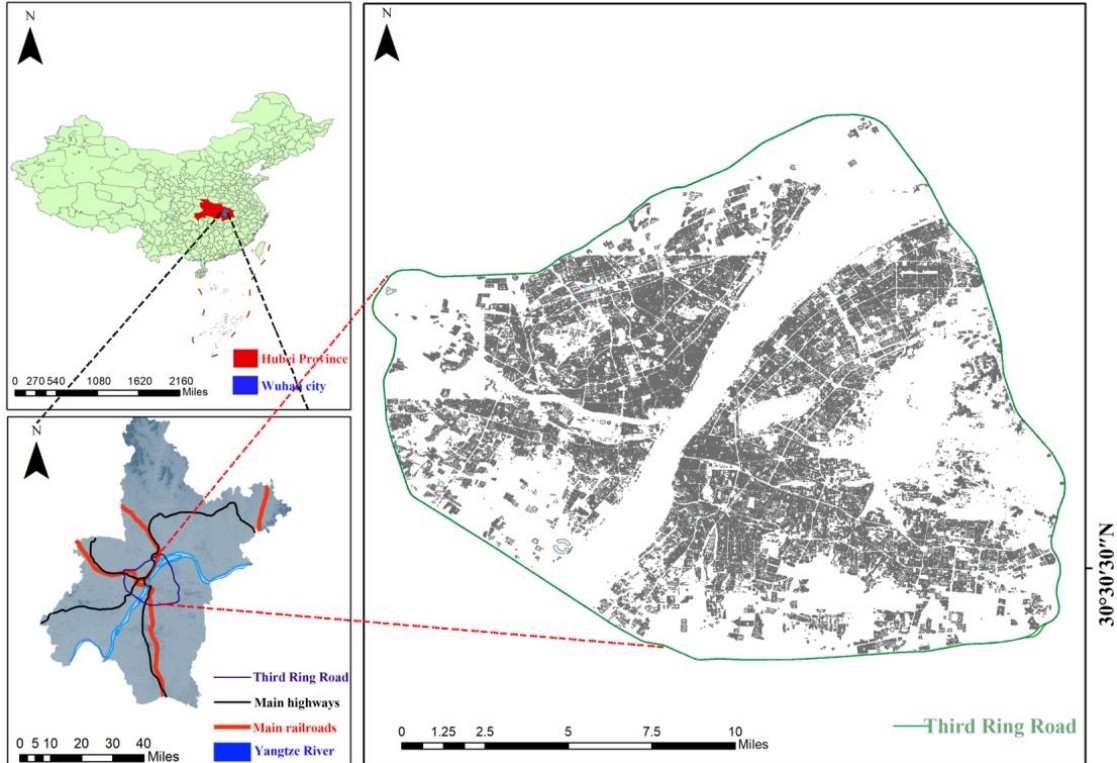

**Figure 1.** Geographic location of the study area: top left geographic location of Hubei Province, China, bottom left geographic location of Wuhan City, Hubei Province, and right geographic location of the study area in Wuhan.

To summarize, we chose Wuhan as the experimental area for this study based on the factors of urban complexity, population diversity, spatio-temporal dynamics, and cultural background.

### 3.2. Methodology

We propose a generalized deep learning framework based on SVIs (Figure 2) to explore the intrinsic connections between different urban elements and human perceptions and the deep relationships between perceptions, which researchers usually neglect. This study attempts to use the semantic segmentation method of SVIs combined with a perception evaluation model to elucidate the weights of the elements in different street spaces and the perception dimensions they affect. This study consists of three main processes: (1) Using Baidu Map's free and open API interface to obtain street view images to construct a Wuhan image dataset. (2) Based on the Wuhan street view image dataset, a deep learning PSPNET model is used to extract the semantic composition of the city streets, and a neural network trained on the place pulse dataset is utilized to quantify the spatial perception evaluation. (3) A multiple linear regression model is established based on the spatial semantic composition and human perception scores. For both the spatial semantic composition and human perception scores, a multiple linear regression model is established to analyze the contribution of each object to the six perceptual attributes.

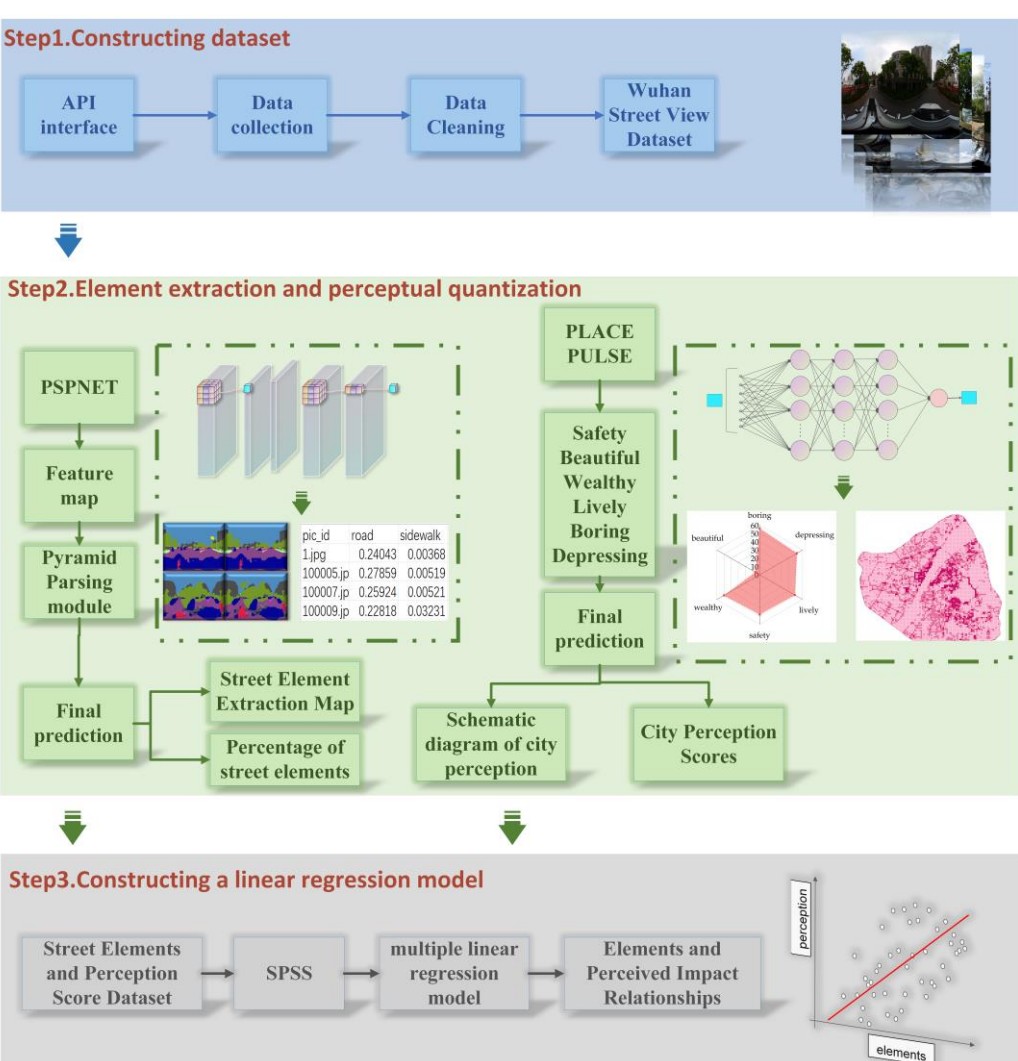

**Figure 2.** Workflow of linear relationship construction between urban spatial elements extraction and urban perception prediction.

### 3.2.1. Constructing Street View Image Dataset

In recent years, SVI data have been widely used in scientific research as proxies for natural reflections and perceptions of the real world. Using openly available SVIs from mapping companies is less time-consuming and labor intensive than traditional on-site data collection or drone photography. SVIs contain a large amount of data covering different geographic locations (SVI services are available for more than 100 countries worldwide) and provide API development interfaces from map providers to access SVIs at the desired location. SVIs contain comprehensive information about urban infrastructure, retrieve images of buildings at the street level, and provide an intuitive and correct representation of urban façade information. Commonly used SVI data sources in the literature are Google Street View (GSV), Baidu Street View (BSV), and Tencent Street View (TSV). The SVI data in this study were obtained from the Baidu Street View database.

This paper selected the city of Wuhan as the research object, considering there is no public, open, accessible Wuhan SVI dataset. We obtained 126,338 Baidu SVIs from 2020 to present (Figure 3) to create a Wuhan SVI dataset for urban street spatial perception experiments. The distances between two neighboring image locations were 8–20 m. Each SVI was labeled with geographic information. This included the latitude and longitude of the location where the image was taken, the azimuth of true north in the picture, the shooting pose information of the image, and the unique identifier of the neighboring streetscape. To improve recognition accuracy, the SVIs should try to satisfy the four conditions of sufficient illumination, weak distortion, complete content, and a high image resolution. After data cleaning, the distorted, damaged, or inconspicuous parts were eliminated, and the image size was unified to 2048 × 1024 pixels by the OpenCV 4.2.0 processing software.

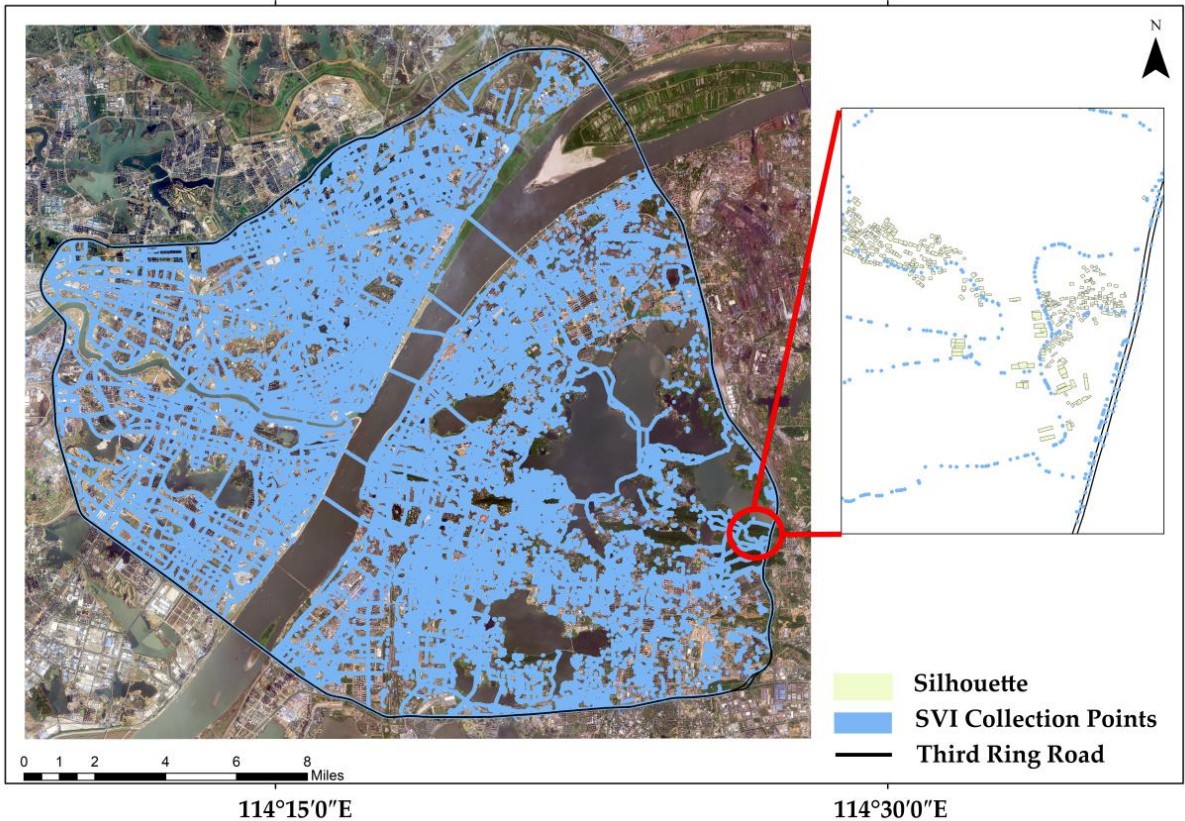

**Figure 3.** Schematic of the study area and SVI collection sites. The distribution of collection points is shown in blue, and the study boundary is in black.

### 3.2.2. PSPNET-Based SVI Extraction Methods

In order to identify the visual elements that may make a place safe, lively, or frustrating, our proposed approach aims to extract the constituent elements in urban street spaces that may be highly relevant to human perception. Recent advances in deep learning have shown that, given an input image, a fully convolutional network (PSPNET) can predict the semantic attributes of each pixel in the image, predict the category labels of each pixel, automatically compute the area occupancy of the semantic objects in the scene, and can be used to generate natural object-level segmentation results.

PSPNET (Figure 4) has been employed explicitly for scene parsing and image segmentation tasks. It was proposed by researchers at the Chinese University of Hong Kong in 2017, aiming to effectively capture multi-scale contextual information in images to improve the accuracy and semantic understanding of image segmentation. CNNs may be limited by the size of the sensory field when dealing with image segmentation tasks, resulting in an insufficient capture of local and global contextual information. This problem inspires PSPNET and aims to enhance the modeling capability of contextual information at different scales by extracting features from different scales by utilizing pyramid structures. PSPNET's core idea is constructing feature pyramids at different scales to obtain multi-scale contextual information. Its core components include a feature extraction network and a pyramid pooling module.

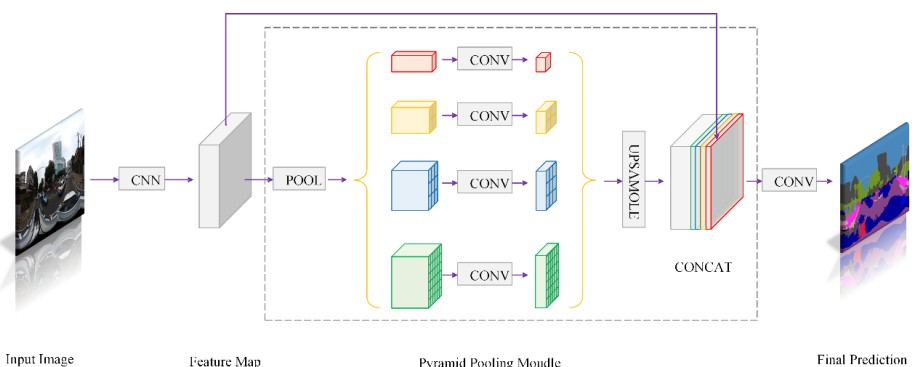

**Figure 4.** Schematic diagram of PSPNET model structure.

Deep CNNs are challenging to train and fit due to vanishing gradients and the curse of dimensionality during the training process. RESNET is considered to be a reasonable attempt to solve this problem. PSPNET uses the existing pre-trained convolutional neural network RESNET101 (Figure 5) as a feature extraction network. These networks extract the bottom- and top-level features from input images. We used RESNET101, which was pre-trained on Places2 [62] an image database containing 10 million well-labeled images.

Compared with other classical semantic segmentation models, such as FCN, PSPNET has several innovations. The most critical feature of PSPNET is that it effectively obtains global contextual information by introducing the pyramid pooling module (PPM). This operation makes PSPNET more capable of understanding complex scenarios with better contextual inferences. Secondly, "Pyramid Pooling" can flatten and stitch together generated feature maps of different levels and scales and input them into the fully connected layer. This global prior module is designed to remove the constraint that CNNs need to input fixed-size images for image classification. Hierarchical global information containing relationships between different sub-regions at different scales is proposed to maintain the contextual information that characterizes the relationships between different sub-regions. Conventional models may ignore objects with small volume features, but too large a volume will exceed the model's range, eventually leading to discontinuous predictions. PSPNET solves this problem well through pooling operations of different sizes. As shown in Figure 4, the pyramid pooling module has four different sizes of pooling operations, $1 \times 1$, $2 \times 2$, $3 \times 3$, and $6 \times 6$, to obtain multiple sizes of feature maps and then

perform the "1 × 1 Conv" operation on the feature maps again to reduce the number of channels. Bilinear interpolation is used for "UPSAMPLE" to obtain feature maps of the same size in front of the pyramid module, which is spliced over the channels. The output of this pyramid pooling module is finally used as the final feature map of the deep neural network and is referred to as the global scene prior information.

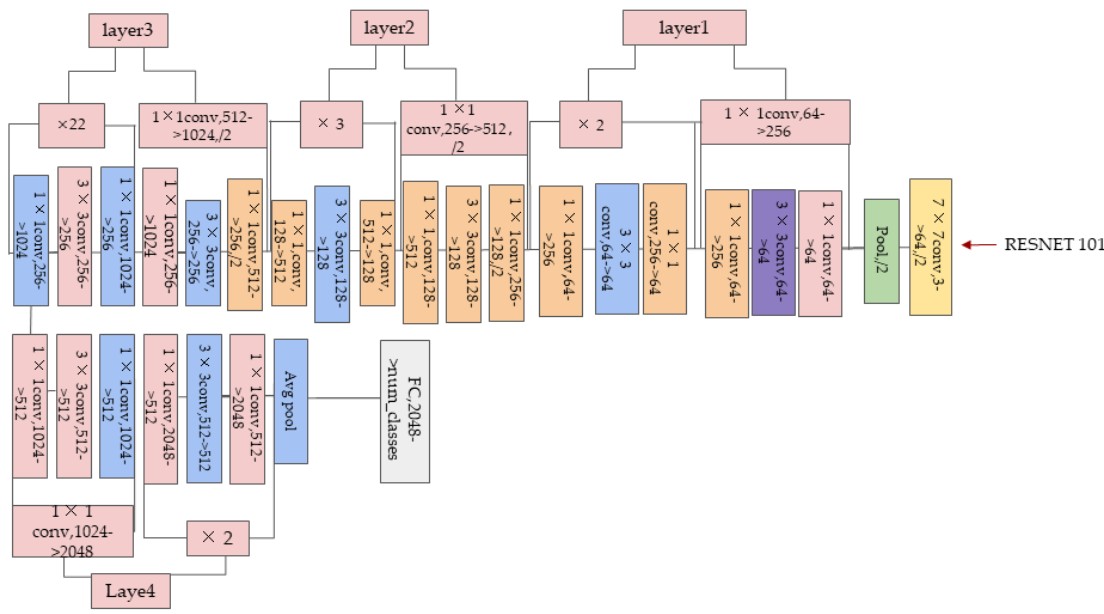

**Figure 5.** Structure of the RESNET101 model.

In recent years, the PSPNET model has achieved significant performance gains in various image segmentation tasks and has a wide range of applications in geological detection and usage, autonomous driving, facial segmentation, and precision agriculture. However, there needs to be more research on applying the PSPNET model to urban street spatial perceptions. In recent years, many scholars have used the PSPNET model to segment and extract the components of cities to quantify the green visibility, sky visibility, road accessibility, and other indexes in these cities. In this study, the PSPNET model was selected as the primary tool for segmenting urban elements, based on which, a new research idea of urban spatial perception was explored. The results of the study encourage developers and city managers to pay more attention to the impact of the environment on the physical and mental health of human beings, which is of reference value for the planning of new development areas, as well as the renovation of old neighborhoods, and promotes the development of the sustainability and livability of urban communities. Therefore, this paper selected PSPNET to segment the urban elements in streetscape images.

### 3.2.3. Training Perceptual Models with the "Place Pulse" Dataset

This study focused on six categories of urban perceptions: wealthy, safety, lively, beautiful, boring, and depressing. These six categories of perceptions were created by annotating and training street scenes from most cities worldwide and voting on two-by-two image comparisons for the final combination. The "Place Pulse" dataset is widely used with SVIs and neighborhood perception information to study the relationships between urban environments and people's perceptions of them. The dataset is designed to help researchers understand the visual characteristics of urban spaces and people's perceptions of different locations.

The "Place Pulse" dataset was constructed based on GSVs worldwide and perception data from social media users (Figure 6). The work was initiated in 2010 by the Social Sensing team at the MIT Media Lab. The researchers built it from more than 110,000 SVIs of city streets in the U.S. and other countries and perceptual data extracted from social media

such as Twitter and Instagram. The dataset possesses a breadth of distribution (containing 56 cities in 28 countries around the world), diversity of volunteer characteristics (across age, ethnicity, and geography), and consistency of scores (no significant differences between attribute groups). Regarding city size, metropolitan areas such as New York and London were included in the dataset, as were Glasgow and Gaborone. For each city, the locations were randomly and densely sampled from the spatial area of the town. Metadata for these images are also included in the dataset, including geographic coordinates and camera heading degrees. Therefore, we consider the dataset to be representative of the perceptions of street space in most cities.

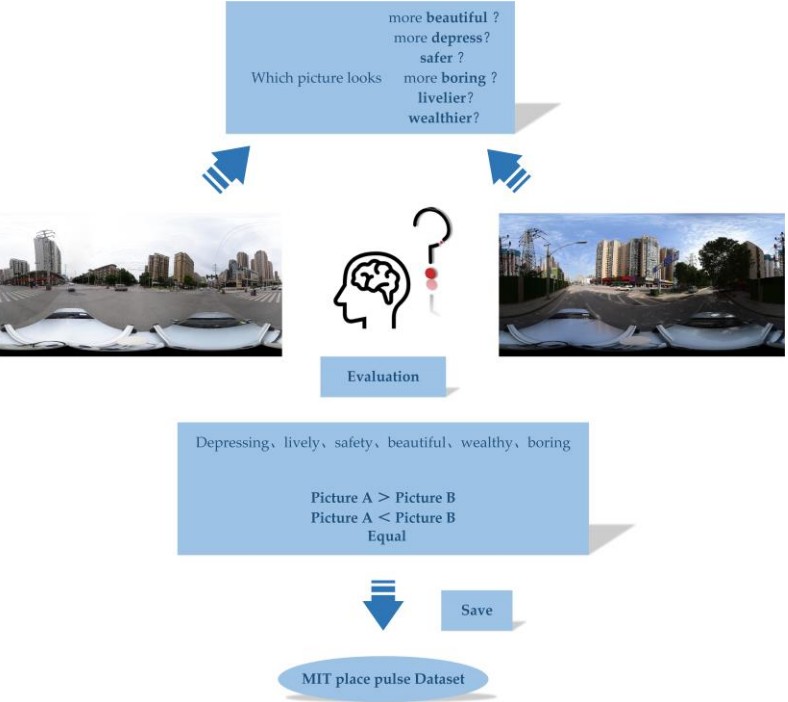

**Figure 6.** MIT "place pulse" dataset construction flowchart.

The dataset contains many SVIs covering a wide range of types of urban environments around the globe, from bustling commercial districts to quiet residential neighborhoods. Each SVI is accompanied by participant-supplied labeling data on landscape perceptions, which can cover multiple perceptual dimensions such as "safety", "beautiful", and "lively".

There are two primary forms of quantitative scores for street perceptions [63,64]. One is based on the "intensity" of comparisons between images. The other is a direct evaluation of individual samples. We adopted the second approach. In real life, it is more practical to directly evaluate the perception of a single sample rather than to compare two scenes, because we would prefer to understand the perception of a scene rather than to compare it with other scenes.

We rated each image by the number of times it was selected and corrected it by the ratio of "winners" to "losers" for all images in comparison. This correction allowed us to adjust the "strength of progress", which we define as the win (*W*) and loss (*L*) rates of image "*s*" over question "*u*":

$$w_{s,u} = \frac{w_{s,u}}{w_{s,u+l_{s,u}+t_{s,u}}} \tag{1}$$

$$L_{s,u} = \frac{L_{s,u}}{w_{s,u+l_{s,u}+t_{s,u}}} \tag{2}$$

where "$W_{s,u}$" and "$L_{s,u}$" denote the number of times image s was selected or not selected in the comparison, and "$t_{s,u}$" denotes the number of times image s was considered to be

equal to another image. Therefore, we can define the Q-value of image s along a particular perceptual metric as:

$$Q_{s,u} = \frac{10}{3}\left(W_{s,u} + \frac{1}{n_s^w}\sum_{j_1=1}^{n_s^w} W_{j_1 u} - \frac{1}{n_s^l}\sum_{j_2=1}^{n_s^l} L_{j_2 u} + 1\right) \tag{3}$$

where "$n_w^s$" equals the total number of images I prioritized, and "$n_l^s$" equals the total number of images I did not prioritize. The win rate of an image "$W_{s,u}$" was corrected by summing the selected images' average win rate and subtracting the selected images' loss rate.

The "Place Pulse" dataset has prompted a large number of studies on the relationship between urban environment and perception [43,65]. Researchers have used the dataset to make many exciting discoveries, for example, that the perceptions of different neighborhoods may be influenced by architectural style, sidewalk width, and vegetation cover. In addition, the "Place Pulse" dataset provides valuable insights for urban planners, designers, and policymakers to better understand how people perceive urban spaces and to guide urban development.

In conclusion, the "Place Pulse" dataset provides an essential resource for studying the relationship between the urban environment and human perception. It promotes deeper thinking and innovation in urban design and planning. Therefore, we chose the "Place Pulse" dataset to train a deep learning perception model to complete our urban street space assessment experiment.

### 3.2.4. Construction of Multiple Linear Regression Models

Multiple linear regression (MLR) is a statistical technique used to model the relationships between a dependent variable (outcome of interest) and two or more independent variables (predictor or covariates). In our study, MLR was used to examine the effects of the spatial components of urban streets on six types of anthropogenic perceptions, exploring the correlations between data from various spaces. They enabled us to assess the strength and direction of these relationships while controlling for confounding factors. Six sets of independent analysis experiments were conducted, with each of the six dimensions of the anthropogenic perception indicators as dependent variables, and the extracted category of the spatial component object of the street was used as a predictor and a fixed effect in the mixed model. The contribution of each object to a particular perceptual attribute was compared by looking at the standardized coefficient of that object in the regression analysis.

Before conducting an MLR analysis, the following assumptions, including linearity, independence, and normality, need to be checked to ensure that the relationship between the dependent and independent variables is approximately linear, that results can be obtained that are independent of each other, and that the residual values conform to a normal distribution. We usually use evaluation metrics to test the goodness of fit of MLR models, such as $R^2$, adjusted $R^2$, VIF, and D-W values, etc. These metrics help us to understand how well a model explains the variance of the dependent variable and whether it is an improvement over the simple model.

In this study, we used the "SPSS" software 27.0.10 to build a multiple linear regression model of spatial constituent elements and human perceptions of urban streets. "SPSS", developed by IBM, is a comprehensive and widely used applied statistical software that provides a versatile set of tools for data analysis, transformation, and presentation. It plays a crucial role in our research methodology.

## 4. Results

### 4.1. Quantitative Results of Urban Perception

A perceptual prediction model pre-trained on the "Place Pulse" dataset was used to simulate the perceptual distribution in the Wuhan region.

This experiment calculated the six categories of perception scores for all the street spaces in Wuhan. Figure 7 shows the randomly sampled image samples and their perceptual scores in the six perceptual dimensions. Then, we took the urban space as a visualization unit and constructed 8054 urban grids of 30 × 30 m. Figure 8 shows Wuhan's six categories of perception mapping, which can be regarded as the "Wuhan Perception Quantization Schematic". In general, the most intuitive conclusion from Figure 8 is that the city center had more positive perceptions than the area around the third ring road, in terms of whether it was "wealthy", "safety", or "lively".

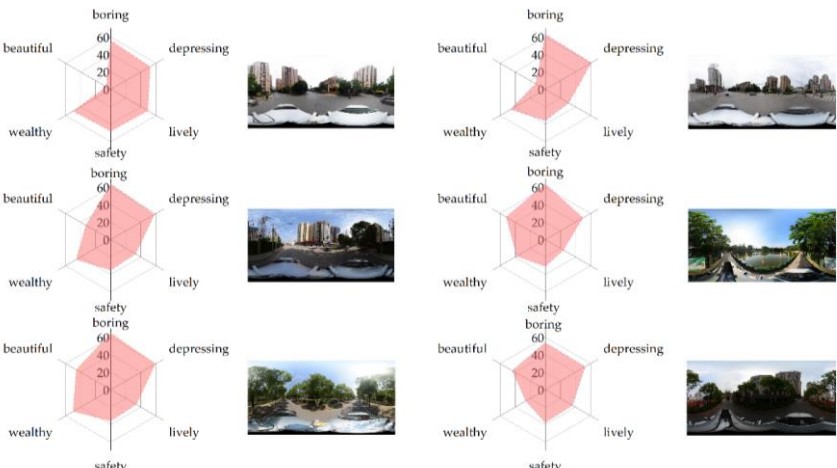

**Figure 7.** Scene perception score graph. The scene perception scores on the left and the street view images on the right were detected for each of the six scenes: "beautiful", "boring", "safety", "lively", "depressing", and "wealthy".

The "Wuhan Perception Quantitative Schematic" shows that the East Lake Scenic Area had the most outstanding beauty perception score, except for the Wuhan Riverfront area. The most famous part of the East Lake Scenic Area in Wuhan is the East Lake Greenway. The East Lake Greenway was the first 5A-level scenic greenway in an urban area in China and has been listed by UN-HABITAT as a "Model Project for Improving Public Space in Chinese Cities". Many famous scenic spots are along the greenway, such as Chu Market and the Li Sao Monument. The greenway is rich in Chu style and Han rhythm. The East Lake Scenic Area effectively combines historical and cultural neighborhoods with modern development. People's perceptions of the East Lake Scenic Area's as "wealthy", "safety", and "lively" were relatively low, and the highest perception of the area was "beautiful".

Although the Hankou riverfront area of Wuhan is slightly less comfortable than the East Lake Scenic Area, it has almost overwhelming advantages regarding education, commerce, medical care, transportation, and other resources. Therefore, in terms of whether it was "wealthy", "safety", "lively", or "beautiful", the riverfront area had a better performance.

We can see from the "Perception Quantification Schematic" that, in Wuhan, "safety" and "wealthy" were highly correlated. Perceptions of "lively" and "boring" showed almost opposite trends. This is consistent with our later conclusion on the influence of urban factors and perceptions, which, to some extent, supports this paper's research. It is not yet known whether the perceptions of all Chinese cities support the above conclusion, and more cities need to be studied in the future to verify this view.

The results of the regional perception study derived from this paper are highly consistent with the economic development level of Wuhan's regions, which, to some extent, confirms the scientificity and effectiveness of the human–machine confrontation model proposed in the study of urban perception.

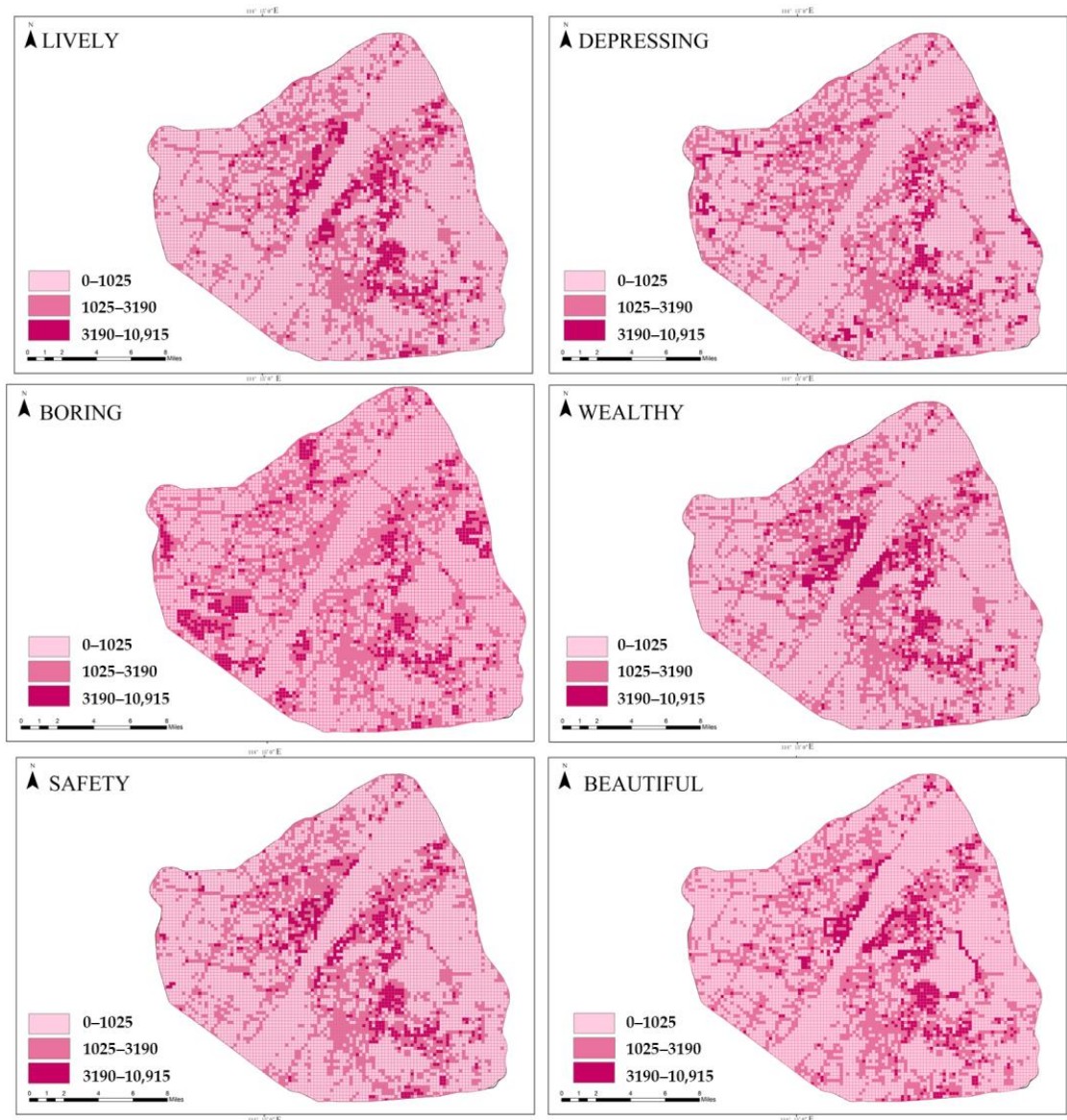

**Figure 8.** Schematic diagram of perceived quantization in Wuhan city.

### 4.2. Semantic Segmentation Modeling Results

In this study, three classical semantic segmentation models were tested for their accuracy using the Wuhan SVI dataset as a test sample, aiming to find the most suitable model for segmenting Wuhan's urban spatial elements. The accuracy of image element recognition, prediction, and extraction was verified and compared by testing the three types of semantic segmentation models: PSPNET, FCN, and U-NET. The three groups of model element extraction results in Figure 9 show the differences in the models' abilities to segment the images' constituent elements. The experimental results of the model performance test show that, compared to the FCN and U-NET models, PSPNET had a better recognition effect for small-volume details such as billboards, pedestrians, or vehicles and is more suitable for urban element extraction work in more complex environments. Moreover, both U-NET and FCN had different degrees of misrecognition, while PSPNET had an excellent performance in this aspect. We also conducted an additional set of comparison tests. Firstly, we selected 500 objects as accurate labels for each spatial constituent element (car, bus, and sky, etc.) and extracted the elements using PSPNET, U-NET, and FCN, respectively. Figure 10 shows the extraction accuracy of each model for the elements. We found that PSPNET had an overwhelming advantage in the element extraction experiments. However,

there was not much difference in the accuracy of extracting the "BUILDING" elements. However, PSPNET showed a better accuracy for more minor elements, such as traffic lights and traffic signs. Based on the average accuracy, PSPNET was more accurate in extracting elements. PSPNET is a more suitable model for urban element extraction. Therefore, we chose PSPNET to complete extracting the urban elements in Wuhan.

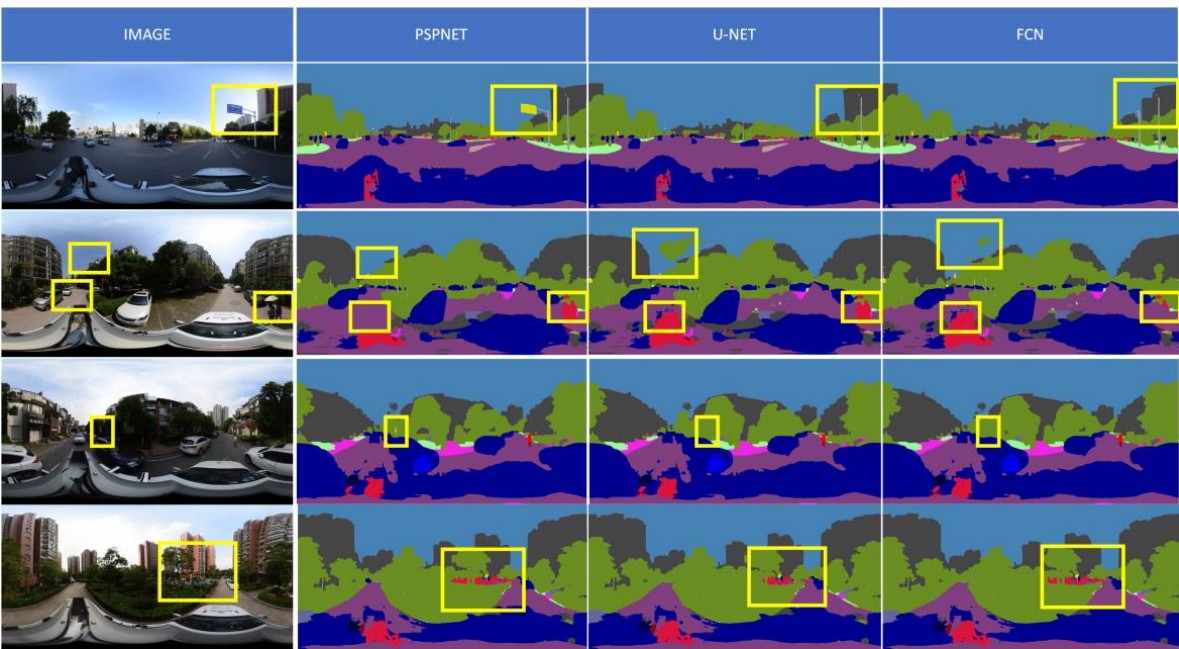

**Figure 9.** Demonstration of three classical semantic segmentation models (PSPNET, U-NET, and FCN) for accuracy testing in Wuhan street view images. In the yellow frame are the different results that each model produces for the element.

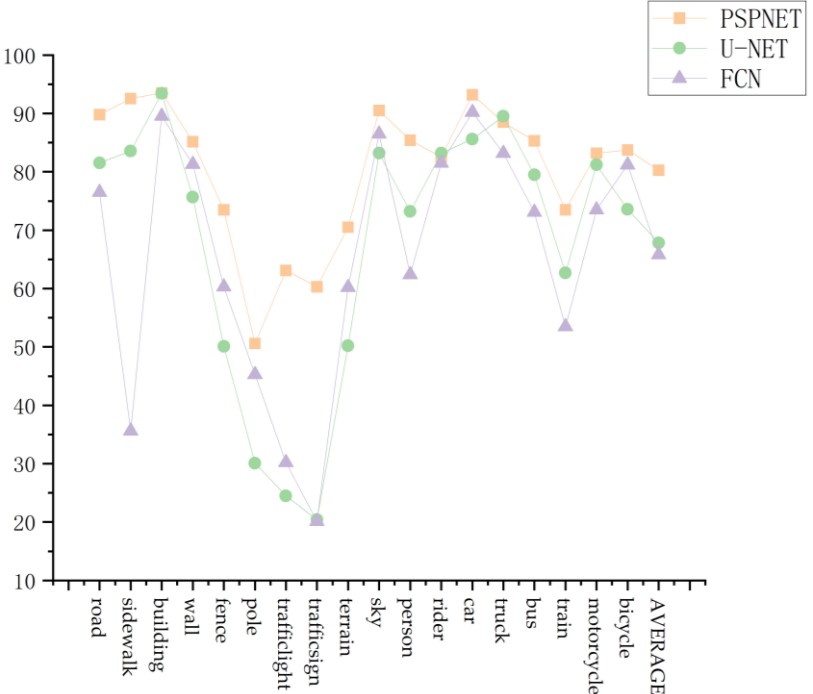

**Figure 10.** Accuracy results of three classical deep learning models (PSPNET, U-NET, and FCN) in street element extraction experiments.

To study the spatial street perceptions at a city scale, we chose the PSPNET convolutional neural network model with the best overall performance among these networks. The hardware setup for testing was as follows: The operating system was Windows 11, and the programming language was Python 3.6. The deep learning platform was Pytorch 1.10.2. The machine learning framework used the scikit-learn library. Image processing was performed using Open-CV. The CPU was an 11th-generation Intel corei7-11800H@2.30 GHz octa-core processor. The GPU was an NVIDIA GeForce RTX 3070 Laptop.

### 4.3. Linear Regression Results

To further investigate the relationship between the constituent elements and human perceptions of street spaces in Wuhan, and to identify the visual factors that may cause a place to be perceived differently, we selected 18 of the most common object categories from the 150 types of urban elements segmented in the SVIs. We tried to establish multiple linear regressions of "street constituents–human perception" with six human perceptions. The results of the multiple linear regression experiments identified the objects that were positively or negatively correlated with the six perceptual indicators and helped us to understand the deeper relationships between the perceptions further.

First, we verified the credibility of the multiple linear regression model of "street constituents–human perception", including a significance test and other indicators (Figure 11), to determine whether the hypothesis of the model of "street constituents–human perception" was valid or not. The model hypothesis of "street components–human perception" was verified. We should note that, with the use of multiple linear regression analysis processes, the significance of the test should include two parts: multiple independent variables and the dependent variable as a whole of the significance of the test (F-test), as well as the significance of the impact of each independent variable on the dependent variable (*t*-test), both of which show the significance of the linear regression test, although the purpose of the test is different.

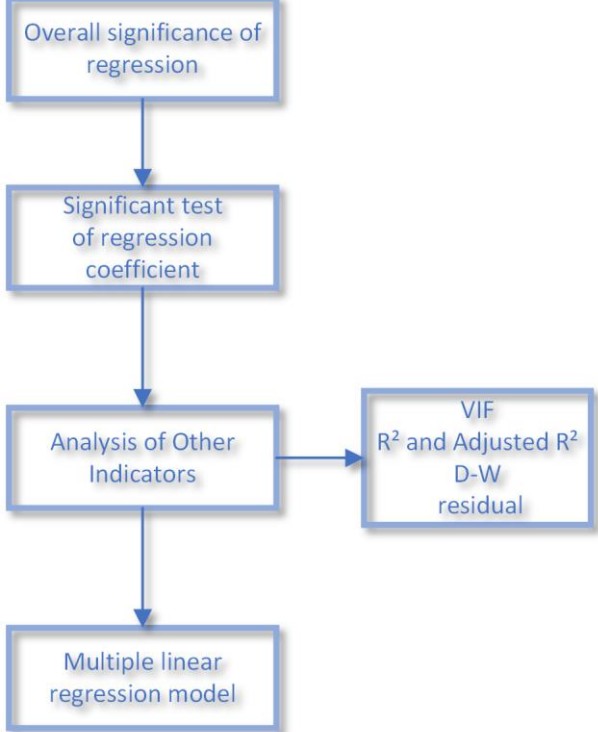

**Figure 11.** Workflow for feasibility analysis of "street constituents–human perception" linear regression models.

### 4.3.1. Overall Significance of Regression

In this paper, we chose the "*F* value" to test the overall significance of the regression model to determine whether the multiple linear regression model was valid. The standard formula for the "*F* value" is (1):

$$F = \frac{\Sigma(\hat{y} - \overline{y})^2 / P}{\Sigma(y - \hat{y})^2 / (n - P - 1)} \tag{4}$$

The "*F* value" test results for the six perceptions of "depressing", "boring", "lively", "beautiful", "wealthy", and "safety" were output as follows (Figure 12). The *F*-values of the statistics were 732.355, 1449.212, 1345.827, 1136.852, 1158.011, and 1346.383, respectively, and the corresponding significance values were less than 0.05, which indicated that, as long as there was an independent variable X (constituent elements), it would have an impact on the dependent variable Y (urban perceptions).

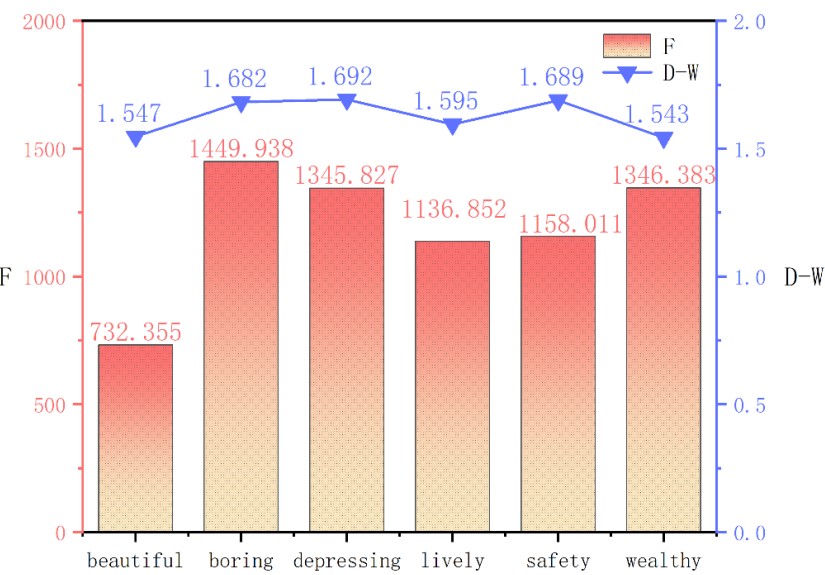

**Figure 12.** The biaxial plot of *F*-value and D-W-value, *F*-values on the left axis and *DW*-values on the right axis.

### 4.3.2. Significant Test of Regression Coefficient

The significant test of regression coefficient refers to the significance test of the effect of each independent variable on the dependent variable, which is carried out using the "*t*-test". The "*t*-test" results of the impact of each independent variable on the dependent variable output from the linear regression model constructed in this paper were as follows: as can be seen in Table 1, the "*p*-values" corresponding to the "*t*-test" of urban elements such as "sky", "wall", and "building" were all less than 0.05, which presents a significance feature. It indicates that all kinds of independent variables X (constituent elements) presented significance to the dependent variable Y (urban perceptions).

**Table 1.** Table of perceived significance of elements.

| Elements | Beautiful | Wealthy | Boring | Lively | Depressing | Safety |
|----------|-----------|---------|--------|--------|------------|--------|
| Road | 0.000 | 0.017 | 0.000 | 0.000 | 0.000 | 0.017 |
| Sidewalk | 0.000 | 0.000 | 0.000 | 0.000 | 0.000 | 0.000 |
| Building | 0.000 | 0.000 | 0.000 | 0.000 | 0.000 | 0.000 |
| Wall | 0.000 | 0.000 | 0.000 | 0.000 | 0.000 | 0.000 |
| Fence | 0.054 | 0.000 | 0.026 | 0.000 | 0.000 | 0.000 |

**Table 1.** *Cont.*

| Elements | Beautiful | Wealthy | Boring | Lively | Depressing | Safety |
|---|---|---|---|---|---|---|
| Pole | 0.000 | 0.000 | 0.000 | 0.135 | 0.0135 | 0.000 |
| Traffic light | 0.025 | 0.000 | 0.000 | 0.501 | 0.0050 | 0.000 |
| Traffic sign | 0.000 | 0.000 | 0.0081 | 0.086 | 0.0086 | 0.000 |
| Terrain | 0.000 | 0.001 | 0.0161 | 0.000 | 0.000 | 0.001 |
| Sky | 0.000 | 0.000 | 0.000 | 0.000 | 0.000 | 0.000 |
| Person | 0.000 | 0.000 | 0.000 | 0.000 | 0.000 | 0.000 |
| Rider | 0.0161 | 0.000 | 0.0301 | 0.000 | 0.000 | 0.000 |
| Car | 0.000 | 0.046 | 0.000 | 0.000 | 0.000 | 0.0066 |
| Truck | 0.000 | 0.000 | 0.000 | 0.000 | 0.000 | 0.000 |
| Bus | 0.000 | 0.000 | 0.000 | 0.000 | 0.000 | 0.000 |
| Train | 0.0088 | 0.000 | 0.003 | 0.000 | 0.000 | 0.000 |
| Motorcycle | 0.000 | 0.000 | 0.000 | 0.000 | 0.000 | 0.000 |
| Bicycle | 0.000 | 0.000 | 0.0401 | 0.0325 | 0.0325 | 0.000 |

### 4.3.3. Analysis of Other Indicators

In this study, in addition to the "overall significance of regression", "significant test of the regression coefficient", "*VIF*" (Figure 13), "$R^2$" (Figure 14), "*D-W*" (Figure 15), and "residuals" (Figure 16) were also analyzed. "*VIF*" is used for a collinearity determination. Collinearity is the correlation between the independent variables with each other that occurs in a linear regression analysis, which significantly impacts the validity of the model and how scientific it is. Collinearity may appear due to a strong correlation between multiple independent variables or an insufficient sample size being collected. In addition, the incorrect use of dummy variables in a regression analysis may also lead to the emergence of the collinearity problem. The standard formula for "*VIF*" is (2):

$$VIF_i = \frac{1}{1 - R_i^2} \tag{5}$$

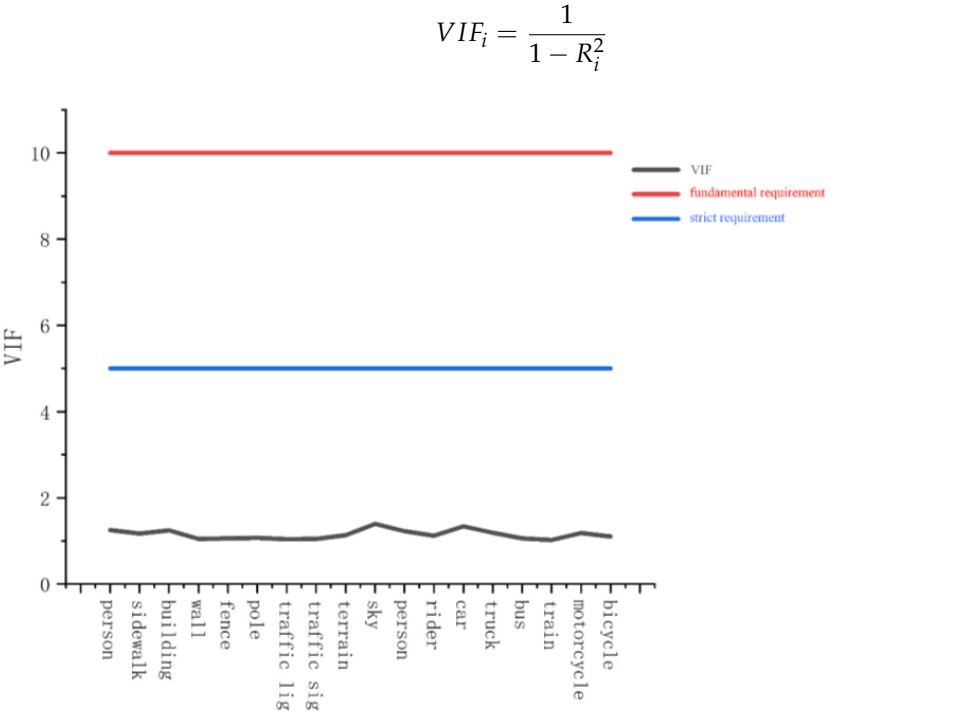

**Figure 13.** The plot of "VIF" for each element.

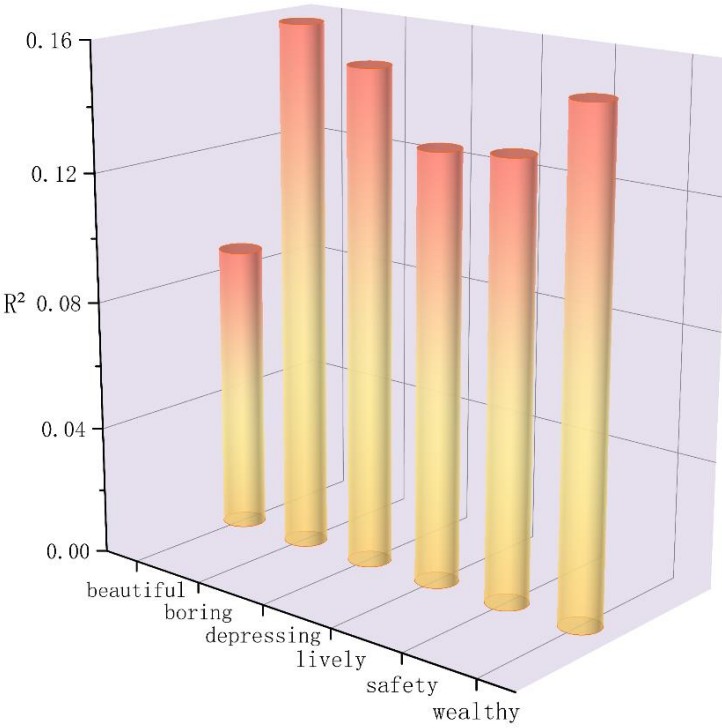

**Figure 14.** Plot of six perceptual "$R^2$". The colors from low to high represent different $R^2$ and each perception is independent of each other.

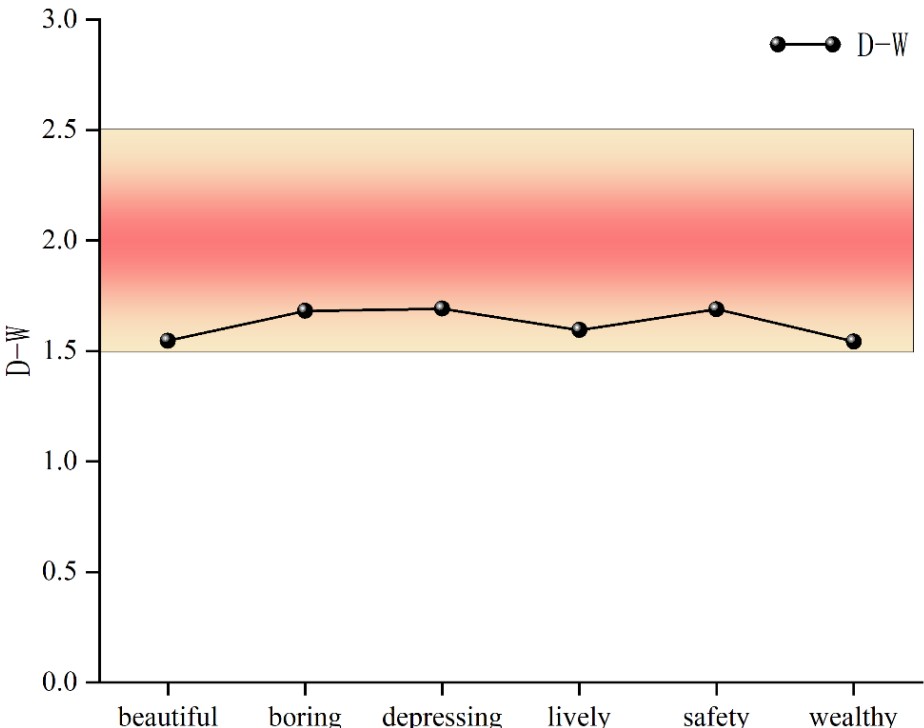

**Figure 15.** Six-perception D-W value plot. The D-W value is as close to the darker color as possible and stays within the interval.

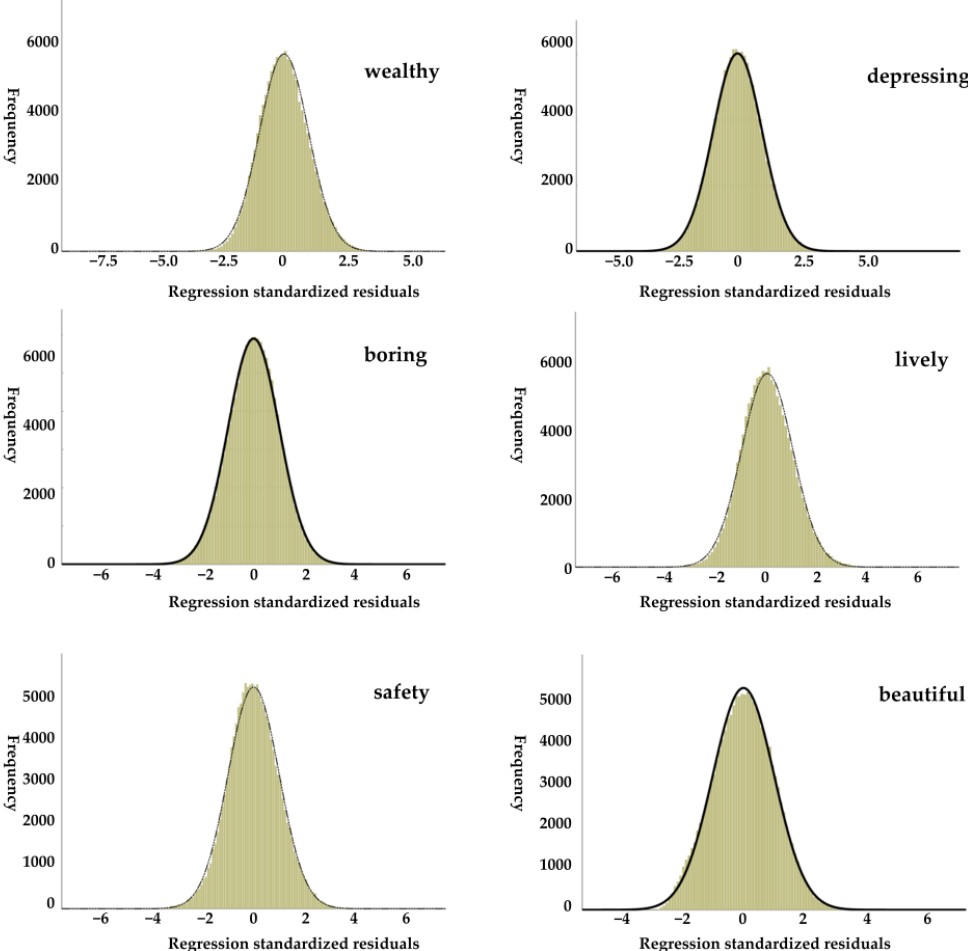

**Figure 16.** Distribution of six perceptual residual values.

A general "*VIF*" greater than 10 (strictly greater than 5) is considered to be severe collinearity. The results of the linear regression models constructed in this study were all much less than 5, so the experimental data selected for this study performed well in the judgment of the collinearity.

$R^2$ is used to analyze the goodness of fit of a model, also known as the coefficient of determination. The value of the $R^2$ ranges from 0 to 1 and represents the degree of fit of a model. Theoretically, the larger the value of $R^2$, the more decisive influence the independent variable has on the results. For example, suppose that $R^2$ is 0.164 in the perceived "boring" study. In that case, the 18 categories of elements selected for this experiment can explain 16.4% of the variation in people's perceptions of "boring". Although 16.4% does not fully explain people's perceptions of the city, it should be considered that we only selected the most representative 18 of the 150 elements extracted by the model for the linear regression hypothesis. Therefore, 16.4% was a satisfactory result. It is worth mentioning that the "beautiful" $R^2$ was only 9.4%. Consider that the assessment of beauty does not depend, to a large extent, on the category of elements in the picture. Different camera angles, photographic-grade compositions, and professional color grading can create different beauty perceptions for the same scene. The standard formula for "$R^2$" is (3):

$$R^2 = \frac{SSR}{SST} = \frac{\sum(\hat{y}_i - \bar{y})^2}{\sum(y_i - \bar{y})^2} = 1 - \frac{\sum(y_i - \hat{y}_i)^2}{\sum(y_i - \bar{y})^2} \tag{6}$$

The multivariate linear regression analysis model established in this paper focused more on the relationship between the influence of urban spatial elements on perceptions. Adjusted $R^2$ is usually used when making model adjustments (increasing or decreasing the number of variables), which is used to judge whether variables should be added to the constructed linear regression model and does not analyze too much.

The random disturbance terms of the model should be independent of or uncorrelated with each other, which is one of the basic assumptions of the multiple linear regression model. The random disturbance term is the error caused by the uncertainty of the data themselves. If the arbitrary disturbance term of the model violates the basic assumption of mutual independence, then the model has autocorrelation. We usually analyze the autocorrelation test using "*D-W*". If the "*D-W*" is near 2 (between 1.5 and 2.5), there is no autocorrelation, and the model is better constructed. On the contrary, if the "*D-W*" deviates significantly from 2, the model has autocorrelation and is poorly built. The standard formula for "*D-W*" is (4):

$$DW = \frac{\Sigma_2^n (e_t - e_{t-1})^2}{\Sigma_1^n e_t^2} \tag{7}$$

Table 2 records the "*D-W*" for the six perceptions. Figure 15 shows that none of the six perceptions deviated from 2, indicating that the model did not have significant autocorrelation.

**Table 2.** D-W values for six perceptions.

| Perceptual | *D-W* |
|:---:|:---:|
| Beautiful | 1.547 |
| Boring | 1.682 |
| Depressing | 1.692 |
| Lively | 1.595 |
| Safety | 1.689 |
| Wealthy | 1.543 |

We analyzed whether the residuals intuitively satisfied normality by examining the normality situation of the residual to judge the model construction. From the six perceptual histograms constructed in this study (Figure 16), it can be seen that the data distribution showed a symmetrical trend in general. In terms of shape, the histograms showed a bell-shaped distribution with "high in the middle and low at both ends". This phenomenon indicates that the research data conformed to a normal distribution, meaning the residuals worked to a normal distribution.

In this study, we analyzed the "overall significance of regression", "significant test of regression coefficient", "*VIF*" (Figure 13), "$R^2$" (Figure 14), "*D-W*" (Figure 15), and "residuals" (Figure 16). From the above graphs, it can be seen that the statistic "*F*-values" were 732.355, 1449.212, 1345.827, 1136.852, 1158.011, and 1346.383, respectively. The corresponding significance values were less than 0.05, so the multiple linear regression passed the test of the overall significance of the regression. The "*VIFs*" were all less than 5, and the "$R^2$" was 1.547, 1.682, 1.692, 1.595, 1.689, and 1.543, respectively. The "*D-W*" did not deviate from 2, and there was no autocorrelation. The "residuals" all showed a normal distribution. This paper's multiple linear regression model of spatial elements and perception has reference value.

### 4.3.4. Analysis of the Relationship between Urban Elements and Perception

Figure 17 presents the multiple linear regression analysis results of "street constituents–human perception". The six human-perceived objects that positively (>0) or negatively (<0) affected the six human-perceived objects were selected and analyzed in depth.

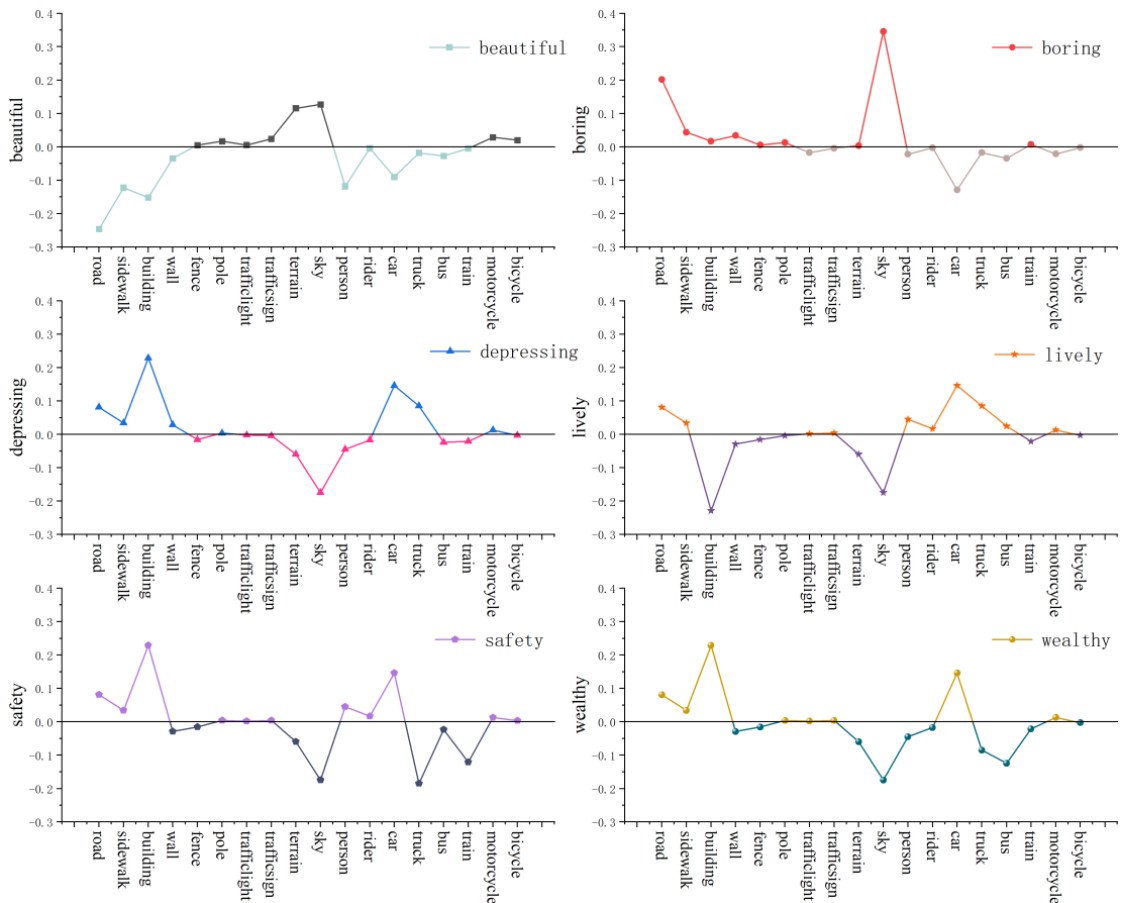

**Figure 17.** Map of urban spatial elements about perceived impacts.

In general, we found a trend of diversification in the objects that contributed significantly to the perceptual indicators. For example, the perceptions of "beautiful", "sky", "terrain", "motorcycle", and "bicycle" were positively correlated with the perception of "beautiful". If the elements "sky", "road", and "sidewalks" appeared more frequently in a scene, it was more likely that people would perceive "boring". On the contrary, "car", "bus", "traffic lights", and "traffic signs" greatly alleviated people's perceptions of "boring" in the scene. We also found that natural elements such as "sky" and "terrain" had a better inhibiting effect on "depressing" emotions. "Pedestrian elements" such as "person" and "rider" also limited the development of "depressing" feelings to some extent. We also found a fascinating phenomenon. If we compare the results of "boring" and "lively", we can see that these two perceptual elements were almost in opposition to each other (Figure 18, left). This is also very much in line with our traditional perception. In a way, "boring" and "lively" are antonyms in themselves.

In the "safety" perception study, we found that, unlike in previous studies, Zhang's analysis [42] suggested that factors such as "car" were positively correlated with the perception of "safety", and we must recognize "truck" and "train" for their efficient workability and transportation capacity, and even "car" for the great convenience it provides to our lives. However, compared to the size of the human race, the above-proposed transportation is too large. Transportation at high speeds can easily cause irreparable damage to cities and pedestrians. From this perspective, "truck", "train", and "car" can hardly be considered as factors that are positively related to "safety". Compared to the mass mentioned above transit vehicles, "motorcycle" and "bicycle" present better "safety" implications.

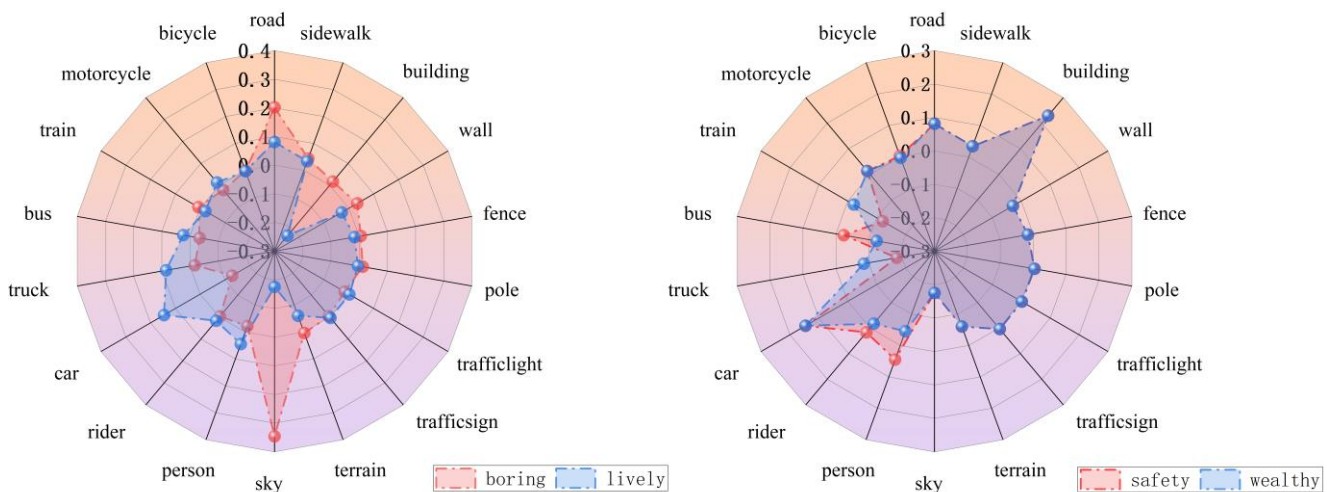

**Figure 18.** Schematic diagram of inter-perceptual relationships. On the left is a schematic of the elements that may have an impact on the perception of "BORING" and "LIVELY", and on the right is a schematic of the factors that may affect the perception of "SAFETY" and "WEALTHY".

In the study of the perception of "wealthy", we found that it was more persuasive to discuss the perceptions of "wealthy" and "safety" together. Figure 18 shows that the urban elements positively correlated with "wealthy" almost also reflected the "safety" level in the scene. The Perceived Confusion Matrix in Figure 19 also shows that "safety" and "wealthy" were also highly correlated, a finding that is consistent with scholars' suggestion that there is a strong relationship between economic development and crime [66]. Economic growth can lead to decreased crime rates, and widening income disparity has always been a significant cause of crime growth, regardless of financial system. The combination of economic and psychological pressures is one of the primary triggers of crime, and regional economic disparities can affect the distribution of crime and increase crime rates. From this point of view, affluent places are indeed more secure than less economically developed ones.

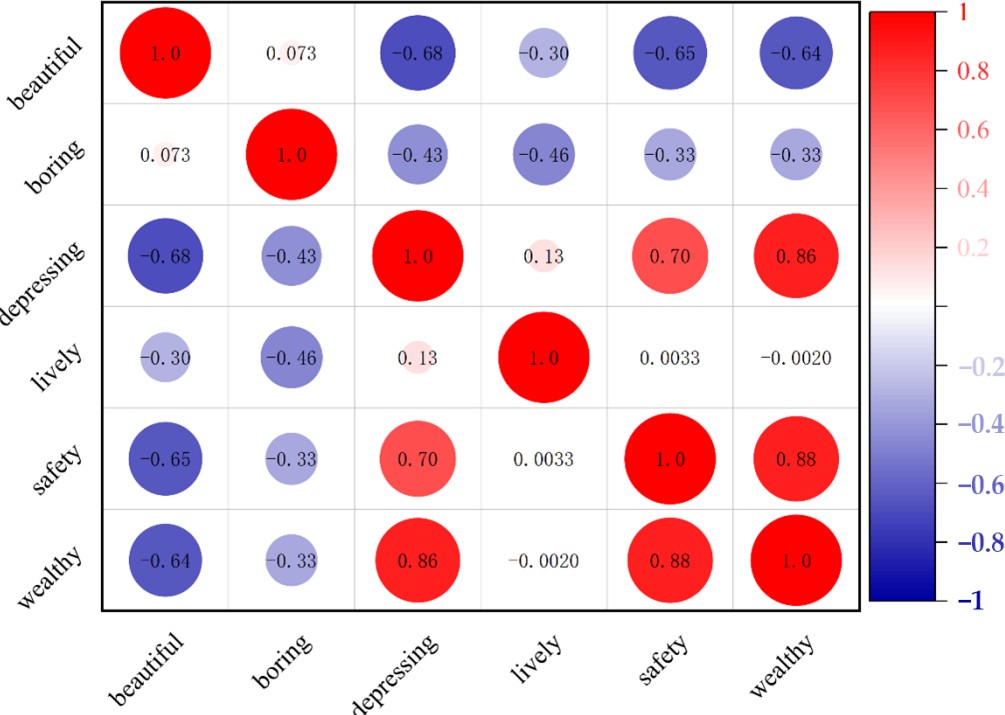

**Figure 19.** Six-perception confusion matrix map.

## 5. Discussion

Street spaces are an integral part of cities, and their visual and spatial quality represents a town's image, affects its inhabitants' living conditions, and may have positive or negative social impacts, which are crucial for the perception and control of the city.

More and more fields have been focusing on human and space perception research. The contributions of this study are as follows: Firstly, this study quantified six kinds of human perceptions of place in Wuhan city based on the deep learning of SVIs. Second, this study tried to analyze the connections between the physical composition of street spaces and the human perceptions of the places, and to analyze the relationship between human perceptions.

Through the study of street space perception, we found that "dynamic elements" similar to "car", "bus", and "person" had a positive effect on the perception of "lively". On the contrary, "static elements" such as "sky" and "building" played negative roles in "lively" perceptions. The above findings are consistent with those of Zhang's study [42], which confirms the study's credibility. However, there are different results from previous studies on the effect of large vehicles on perceptions of "safety", so we need to continue to validate our findings in future studies on other cities. The impact of the environment on physical and mental health should also be emphasized. Marco Helbich et al. [57] verified the preventive effect of blue-green space on depression in Chinese seniors and confirmed the feasibility of the deep learning of street view data for health-related assessments of automated environmental exposure.

We found that "car", "building", "road", and other "urban products" were highly representative of the affluence of urban space. The development of urban roads can enhance the openness of the economic system, and good transportation conditions can accelerate the economic cycle within a region and promote economic development. Previous studies have also confirmed this view. We also found a high correlation between "wealthy" and "safe". Tian [67] used cross-sectional data from 31 provinces in China and 15 years of panel data analyses of 15 provinces to test the statistical relationship between the crime situation and economic growth, mobile population, and urbanization in different regions, and analyzed the impact of regional economic differences on the crime situation with related theories, confirming the direct correlation between economic development and the safety coefficient. However, we still found some problems during the experiment.

### 5.1. Street View Images as a Proxy for Research

First of all, SVIs were selected as the experimental data in this study, which saves a lot of human and material resources compared to traditional research methods and means that urban-level research can be carried out cost-effectively. However, images cannot wholly replace human perceptions; even the most data-rich 360 panoramic images may miss certain information. Strictly speaking, the same person in the same place, different temperatures, weather, and even different times of the day may give people different feelings. Secondly, SVIs focus more on urban areas along streets. Street spaces occupy a vital position in cities but do not represent the whole city. Urban places beyond streets, such as campuses, alleys, and historic districts where vehicles are not permitted, are also part of cities, and perceptual studies of these areas should not be neglected. Updating street view images should also be taken into consideration. Usually, the urban areas of first- and second-tier cities are updated once every two to three years, such as Beijing, Shanghai, Guangzhou, and other developed areas. However, some underdeveloped areas still need to start updating the street view images of 2013. Outdated streetscape images can hardly reflect the changes in the urban environment in recent years, which is not conducive to urban spatial research, results in a lack of timeliness of experimental results, and is usually accompanied by significant errors.

### 5.2. "Perceived Bias"

A fully automated scoring system based on deep learning provides excellent convenience for our research. Of course, no algorithm can achieve 100% accuracy, and we needed

to calculate the algorithm's accuracy manually. Therefore, we selected 2000 "wealthy" streetscape city areas, combined with the economic development data of the local area of the streetscape to score them manually, and labeled them as real labels to compare them with the scoring of the deep learning model. Figure 20 shows a picture of the randomly selected Wuhan SVI dataset, with predicted affluence scores increasing from left to right. The predictions matched our intuition that areas with wide, smooth roads and neat buildings tended to be more "wealthy", while areas with awnings, dirty fences, and dirt roads were considered to be less "wealthy".

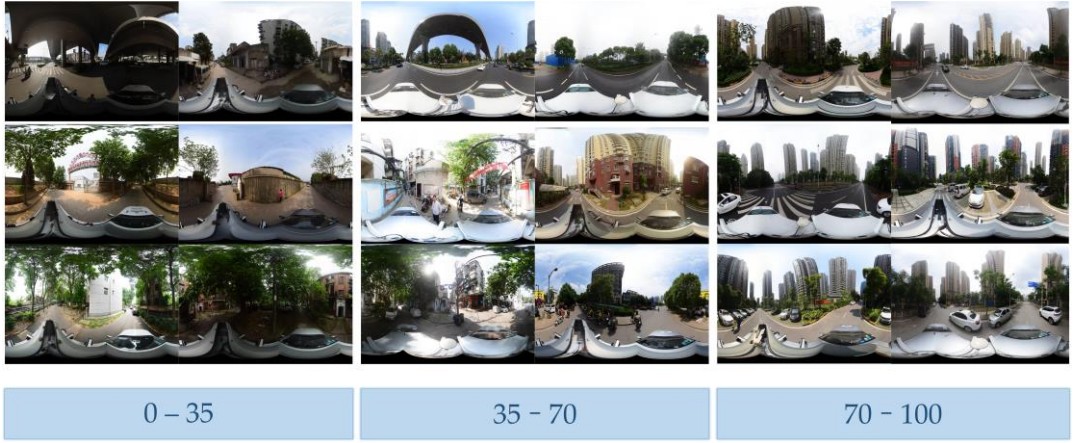

Figure 20. Low to high scores of "wealthy" images from the Wuhan SVI dataset.

However, through a manual review, combined with the data of Wuhan's regional economic development level, we also found a small amount of "perceptual bias" phenomena. These can be categorized into two main types: scored lower than or higher than the actual score by the deep learning model (Figure 20). For the perception of "wealthy", the evaluation of the wealthiness of the scene through the SVI mainly considered the composition of the elements, the urban environment, and the degree of cleanliness. It is undeniable that these factors had a significant impact on the region's "wealthy" indicators. However, individual elements, such as geographic location, cultural, and historical values, cannot be assessed solely based on images.

By analyzing these two "perceptual biases", we found that the deep learning model scores were lower than the actual scores, mainly due to the following reasons. We found that the areas where perceptions of "wealthy" were much lower than the actual score were primarily concentrated in historic districts or traditional buildings in the city center that had not been demolished. However, the environment of these districts is hardly similar to that of an upscale neighborhood or economic center from the outside. However, they represent the comprehensive cultural heritage of the city's region and era, are a humanistic reproduction and spatial and temporal marking of the city's traditional living scenes and the cultural atmosphere of the period, are the oldest and most authoritative witnesses of the city's age, and are the most eye-catching and typical symbolic image given by the long history and culture of the city.

In addition to the above, there were a few districts where the score of "wealthy" was much higher than the actual score. The case studies showed that some areas have beautiful environments, clean streets, and good facilities. However, most of these areas are far from the city center, and accessibility to education and healthcare still needs to be further improved, so the predicted scores of these areas were higher than the actual scores. The above reasons made the deep learning model have "perceptual bias".

Research involving human perceptions has been a worldwide challenge, and it is difficult to fully explain the feelings an environment inspires in humans using only image data, because the human senses go far beyond vision. For example, in this study's phenomenon of affluence bias, we did not consider factors such as residents' income, population loss,

regional housing prices, and transportation. The psychological and physical effects of affluence on residents' daily lives should also be the focus of research. In future research, we should consider more influencing factors. These are not only based on computer vision, but also on cultural experience, history, and social development, which are difficult to express in graphic data, but are closely related to the human senses (Figure 21).

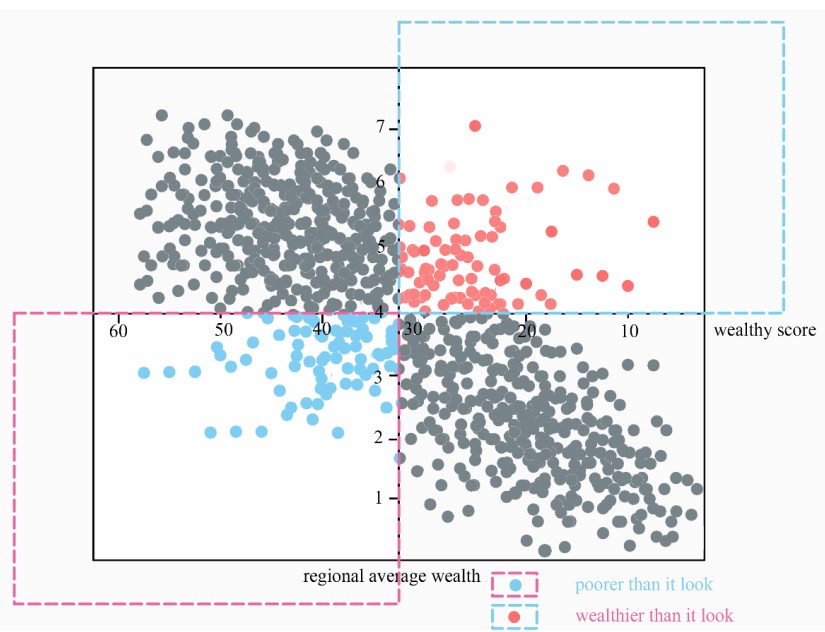

**Figure 21.** Wealthy sensory difference median matrix. The red part of the blue box is "wealthy", which is higher than the model score, and the blue part of the red box is "wealthy", which is lower than the model score. The gray portion is "wealthy" and matches the model score.

## 6. Conclusions

Perception studies of urban street spaces and exploring the street space elements that influence perceptions have been a matter of interest in geographic information and urban science. However, measuring the human perceptions of spatial environments at the metropolitan level has been challenging due to insufficient data and appropriate processing methods. In this paper, we applied the latest deep learning techniques combined with SVIs to realize a study of street space perceptions in Wuhan and explore the effects of the common elements in the city on perceptions and the deep relationships between these perceptions. First, a Wuhan SVI dataset was established to address the lack of publicly available image data in Wuhan. Second, we proposed a spatial perception analysis method based on the deep learning of SVIs. We used the Perception Prediction Model to quantitatively analyze the street spaces in Wuhan. The PSPNET model was used to segment the components of the Wuhan SVIs. Finally, we analyzed the multiple linear regression analysis between the perceived situation of the city and the natural spatial elements to determine the factors that affected human perceptions.

In this paper, we proposed a research idea different from the traditional one for spatial perception quantification based on SVIs and deep learning technology, which solves the problems of being time-consuming, subjective judgments, and difficulties in being promoted on a large scale in the previous work. We studied three classical deep learning frameworks (PSPNET, FCN, and U-NET) and finally selected the PSPNET model to extract the urban spatial elements of Wuhan. Our research deepens the understanding of how urban spatial perceptions are formed and quantified, delves into the relationships between characteristic features in urban environments and the human perceptions of urban spaces, and contributes to the field of urban planning and regeneration by demonstrating novel applications in the context of big data and artificial intelligence. The results can provide urban planners with the tools to assess and improve the quality of urban spaces based on

public perceptions. This would enhance the tourist experience by identifying attractive urban features and the areas that may need improved safety measures based on their perceived safety. This would help to improve the quality of cities and bring about tangible benefits for policymakers, planners, businesses, and residents. This is essential for creating urban environments that are more livable, sustainable, and responsive to people's needs.

**Author Contributions:** Conceptualization, H.X. and H.S.; methodology, H.X. and H.S.; software, H.H. and H.S.; validation, H.H. and H.S.; formal Analysis, H.X. and H.S.; data inspection, H.S., Q.W., Y.Y., Z.C., X.L., J.Z. and T.L.; writing—original draft preparation, H.X. and H.S.; writing—review and editing, H.X.; visualization, H.X. and H.S.; supervision, H.X.; project administration, H.X.; funding acquisition, H.X. All authors have read and agreed to the published version of the manuscript.

**Funding:** This work was partly supported by the National Natural Science Foundation of China (41771473), the Hubei Changjiang National Cultural Park Construction Research Project (HCYK2022Y20), the Hubei Construction Science and Technology project ([2022]2198-123) and the 2021 Construction Science and Technology Plan Project of Department of Housing and Urban-Rural Development of Hubei Province (No: 202115).

**Institutional Review Board Statement:** Not applicable.

**Informed Consent Statement:** Not applicable.

**Data Availability Statement:** Not applicable. Data are processed by architectural professionals and can be contacted if needed.

**Conflicts of Interest:** The authors declare no conflict of interest.

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
