# Peer review of "A Spatial Analysis of Urban Streets under Deep Learning Based on Street View Imagery: Quantifying Perceptual and Elemental Perceptual Relationships"

_sustainability, doi:10.3390/su152014798_

Round 1

Reviewer 1 Report

This study investigated the urban street perception with deep learning based on street view imagery. The six perceptions of large-scale urban areas included safety, lively, beautiful, wealthy, depressng and boring. It is interesting and helpful for the research of measuring human perception of urban street space and exploring the street space elements.

However, there are some issues or questions could be addressed:

1) In the abstract, the important meaning and quantitative results of the experiments in this study should be explained clearly. What are the novel and crucial contributions in this study?

2) In the Section 1 of Introduction, the important meaning and existed problems of the urban street perception should be indicated in detail. Then, the authors should explain the research objectives.

3) In the Section 2 of Literature Review, the explanations of existing studies and their grouped subsections should be more logical. Additionally, the problems of previous studies or the challenges should be summarized clearly. 

4) In Section 3.2 of Methodology, the key framework of this study and following subsections should be consistent, and the detailed subsections should be the logically procedures, not a simple processing. 

5) The Section 3.2.2 and its novel contributions should be explained clearly. What are the differences between this study and PSPNET?

6) In Section 3.2.3, why were the training perceptual models trained with the "Place Pulse" dataset? How was the trained deep learning network applied in Wuhan city? What are the differences with the network trained directly with datasets from Wuhan?

7) In figure 6, what are the meanings for the scene perception score graphs? One image should have one type of scene perception, or many types, as results showed the possible probabilities.

8) In figure 7, the schematic diagram of perceived quantization should be one map, including the six scene perception. Did it should one place indicated one type of urban scene perception.

9) In Section 4.2, is there the quantitative result for the comparison between PSPNET, FCN, and UNET? The quantitative result could indicate the better performance of PSPNET.

10) In Section 4.3, it should be indicated in the main framework of Section 3 of Methodology, as it investigated the independent issue about the relationship between urban elements in the SVIs and the human perceptions.

11) The Section 4.4 could be combinated with Section 4.3.

12) Some details and spelling in the manuscript need to be carefully checked, such as Chinese words in figure 9, the title of Section 3.1, etc.

13) In Section 5 of Discussion, the disadvantages of this method should be deep explored and compared with other methods.

14) Some related studies should be included and cited in this study, such as:(1) Wang, et al, 2022, https://doi.org/10.3390/rs14020265; (2) Li, et al, 2023, https://doi.org/10.1016/j.jag.2023.103303; et al.

Minor editing of English language is needed.

Reviewer 2 Report

I would like to thank the authors for a very interesting research topic, which is indeed very current in the world. I would like to make some suggestions in order to improve the quality of the manuscript. 

It can be seen that the abstract is very detailed and I would not suggest changing it, because it contains all the important data related to the results and the goal of the research. The introductory part is fully adequate and meets the standards, as it provides the necessary elements of the introduction to the research problem.

A very commendable part of the review of literature dealing with current issues. The methodology is described very precisely and in detail in the spirit of understanding the wider mass of readers. The applied methodology gives indications that the authors used a common model in solving the problem, and presented the results through excellent figures.

I would ask the authors to strengthen the chapter related to the conclusion, where they would clarify much more what the theoretical and practical implications of the research and results are. Also, I would ask the authors to strengthen the references and arrange them in accordance with the technical instructions of the journal. I offer one reference:

Risks in the Role of Co-Creating the Future of Tourism in "Stigmatized" Destinations. Sustainability 2022, 14, 1-19, 15530

Reviewer 3 Report

I am assigned to review the paper titled 'Spatial Analysis of Urban Streets under Deep Learning based on Street View Imagery: Quantifying Perceptual and Elemental Perceptual Relationships'. The paper is lengthy however is very interesting. It can be accepted after the following changes:

Comment 1: The quality of Figure 1 is needed to be improved. The scale in the first part of the figure is not visible. The blue color of the legend seems very light.

Comment 2: The text is not visible in the figure 2.  The text on the network diagram has a very low font size.

Comment 3: Figure 6 has the same problem. 

Comment 4: In the discussion section, the results should be compared with some standard results of previously published studies. It will enhance the quality of the manuscript.

Comment 5: The formatting of the references seems inconsistent. 

Comment 6: The authors can use any free grammar check software for minor editing.

*******************ALL THE BEST****************************************

Minor editing is required.

Round 2

Reviewer 1 Report

The author has addressed the issues and suggestions related to my last review. I hava no other questions.

Minor editing of English language is needed.